# Haplotype-resolved gapless genome and chromosome segment substitution lines facilitate gene identification in wild rice

Jingfen Huang[1,6], Yilin Zhang [2,3,6], Yapeng Li[4,5], Meng Xing[1,4], Cailin Lei[1,4], Shizhuang Wang[1,4], Yamin Nie[1,4], Yanyan Wang[1,4], Mingchao Zhao[4,5], Zhenyun Han[1], Xianjun Sun[1], Han Zhou[2,3], Yan Wang[3], Xiaoming Zheng[1,4], Xiaorong Xiao[4,5], Weiya Fan[1], Ziran Liu[1], Wenlong Guo[1], Lifang Zhang[1], Yunlian Cheng[1], Qian Qian[1,4], Hang He[2,3] ✉, Qingwen Yang [1,4] ✉ & Weihua Qiao [1,4] ✉

The abundant genetic variation harbored by wild rice (*Oryza rufipogon*) has provided a reservoir of useful genes for rice breeding. However, the genome of wild rice has not yet been comprehensively assessed. Here, we report the haplotype-resolved gapless genome assembly and annotation of wild rice Y476. In addition, we develop two sets of chromosome segment substitution lines (CSSLs) using Y476 as the donor parent and cultivated rice as the recurrent parents. By analyzing the gapless reference genome and CSSL population, we identify 254 QTLs associated with agronomic traits, biotic and abiotic stresses. We clone a receptor-like kinase gene associated with rice blast resistance and confirm its wild rice allele improves rice blast resistance. Collectively, our study provides a haplotype-resolved gapless reference genome and demonstrates a highly efficient platform for gene identification from wild rice.

The domestication of cultivated rice (*Oryza sativa* L.) from its ancestral progenitor wild diploid rice, *Oryza rufipogon*, is considered one of the most important developments in human history, and rice is now a staple food feeding more than half of the world's population[1,2]. *O. rufipogon* has adapted to different ecological environments, spanning a broad geographical range of global pantropical regions, with extensive distribution in diverse natural habitats in southern China[3,4]. During the course of domestication and breeding of cultivated rice, many desirable traits, including abiotic tolerance and biotic resistance, were lost as genetic diversity was profoundly decreased[5]. The genomes of wild rice constitute an important reservoir of agronomic, biotic resistance, and abiotic tolerance traits for rice genetic improvement, and also provide fundamental data with the potential to illuminate plant genome evolution within a short timeframe. Strategies to harness such traits for rice improvement have shown clear promise, as exemplified by the introgression of the wild-abortive sterile gene from *O. rufipogon* for hybrid rice development[6].

The importance of *O. rufipogon* for rice improvement has been well-documented, and substantial progress has been achieved in *O. rufipogon* utilization as genomics and genetics methods have advanced[5,7,8]. Harnessing the genetic diversity harbored by *O. rufipogon* for rice improvement requires: (1) a high-quality reference

[1]State Key Laboratory of Crop Gene Resources and Breeding, Institute of Crop Sciences, Chinese Academy of Agricultural Sciences, Beijing, China. [2]School of Advanced Agriculture Sciences and School of Life Sciences, State Key Laboratory of Protein and Plant Gene Research, Peking University, Beijing, China. [3]Peking University Institute of Advanced Agricultural Sciences, Shandong Laboratory of Advanced Agricultural Sciences at Weifang, Weifang, Shandong, China. [4]National Nanfan Research Institute (Sanya), Chinese Academy of Agricultural Sciences, Sanya, Hainan, China. [5]Hainan Academy of Agricultural Sciences, Haikou, Hainan, China. [6]These authors contributed equally: Jingfen Huang, Yilin Zhang. ✉e-mail: hang.he@pku-iaas.edu.cn; yangqingwen@caas.cn; qiaoweihua@caas.cn

genome, and (2) a suitable and permanent genetic population in a comprehensively characterized genetic background. Assembly of a high-quality reference genome of *O. rufipogon* has proven difficult because of its relatively high heterozygosity. So far, many *O. rufipogon* accessions have been sequenced, including several accessions with chromosome-scale assemblies[3,8–10]. In addition, large (or super) pan-genomic studies of the *Oryza* family have been performed[8,10], in which *O. rufipogon* accessions were de novo sequenced. However, a gap-free genome is needed to better understand the genomic/genetic diversity of *O. rufipogon*, and more representative genomes of high quality from wild rice populations distributed among distinct geographical regions will facilitate the study and utilization of the valuable genetic resources of wild rice.

Chromosome segment substitution lines (CSSLs) are powerful tools for identifying naturally occurring, favorable alleles in the germplasm of the wild relatives of crops[11]. In the last two decades, the utility of CSSLs in the identification of genomic regions and quantitative trait loci (QTL) hot spots influencing a wide range of traits has been well demonstrated in wild rice[12]. However, their incompleteness and fragmentation limit their use for wild rice functional studies. Further efforts are needed to fill the gaps between genomic studies and gene identification, especially for salt tolerance and rice blast resistance-related genes, which are much-needed in current rice breeding and production[13,14].

In this work, we report the haplotype-resolved gapless genome assembly and annotation for an *O. rufipogon* accession by integrating Hi-C, BioNano, Nanopore, and HiFi techniques. Compared with previously assembled *O. rufipogon* genomes, this genome assembly shows considerable improvements in contiguity, completeness, and correctness. We construct two CSSL populations to introduce favorable chromosome segments from *O. rufipogon* into cultivated rice, which allows us to identify QTLs associated with agronomic traits and resistance to biotic and abiotic stresses. These resources will accelerate functional genomic studies of wild rice and facilitate future breeding programs for rice improvement.

## Results

### Morphology and genetic structure analysis of an *O. rufipogon* accession

First, we investigated more than 1000 accessions from their original habitats throughout China, with high geographical and agricultural phenotypic diversity. The biotic stress resistance and abiotic stress tolerance of this panel were evaluated. Accession Y476, which was collected from Sanya Cty, Hainan province had excellent overall resistance to various biotic and abiotic stresses. Y476 has the typical characteristics of wild rice, including creeping and vigorous growth capacity, long awns, purple and completely exserted stigma, black hulls, and reddish-brown pericarps (Fig. 1a). Y476 was immune to specific isolates of *Magnaporthe oryzae* (*M. oryzae*), and its salt tolerance was better than that of the cultivated rice variety Nipponbare (Nip) (Fig. 1b, c).

To characterize the genome composition of Y476, we generated 26.2 Gb paired-end (PE) Illumina short reads (Supplementary Table 1) and performed population structure analysis combined with whole-genome re-sequencing data from 446 *O. rufipogon* accessions, as well as 268 *O. sativa aus*, 1882 *O. sativa indica* and 1194 *O. sativa japonica* accessions[2,15]. We performed phylogenetic analysis and principal component analysis (PCA) of the rice accessions. *Japonica*, *indica,* and *aus* rice formed completely isolated clusters, whereas *O. rufipogon* were classified into three types, Or-I, Or-II, and Or-III, according to the classification by Huang et al.[2] Y476 was among Or-clusters, but not clearly classified into a specific subgroup (Supplementary Fig. 1a, b). We further analyzed the admixture pattern of Y476 to confirm its genomic composition using the program ADMIXTURE. With a K-value of 6, six genome types were identified as Or-I, Or-II, Or-III, *aus*, *japonica*-type, and *indica*-type. Y476 was found to comprise 69.5% of the

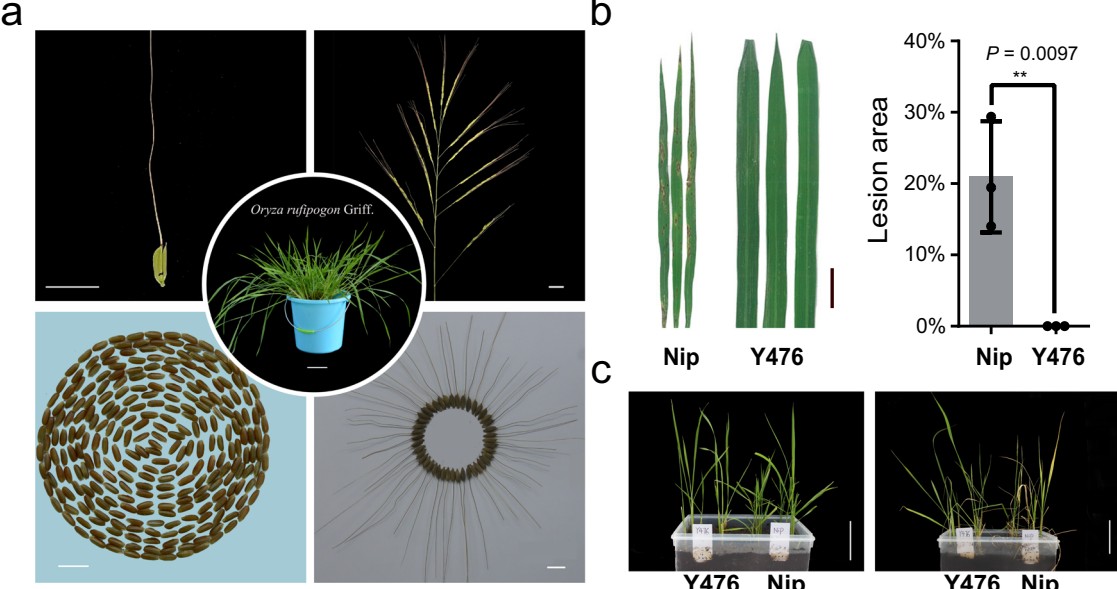

**Fig. 1 | Morphology, rice blast resistance and salt tolerance of the *O. rufipogon* accession Y476. a** Plant type, seed, panicle, spikelet, and caryopsis morphology of Y476. The images show typical wild rice traits, such as creep growth, long awns, purple stigma, spreading tillers, black hull color and reddish-brown pericarp color. The scale bar corresponds to 10 cm in the middle photo and 1 cm in the others. **b** Y476 was much more resistant to rice blast in comparison with Nip. Plants were inoculated with blast isolates FJ07-5-2 and FJ07-8-1. Leaves were collected from Y476 and Nip for measurements of lesion area. Y476 was nearly immune to rice blast *Magnaporthe oryzae* isolates. Scale bars = 1 cm. Results are presented as the mean ± SD from three biological replicates (*n* = 3). Comparisons were performed by two-tailed Student's *t* test (**P* < 0.05, ***P* < 0.01). **c** The salt tolerance of Y476 was significantly higher than that of Nip. The photo shows Y476 and Nip after treatment with 150 mM NaCl for 15 days. Scale bars = 10 cm. Source data are provided as a Source Data file.

**Table 1 | Comparison of the Y476 genome with previously published assemblies of wild and cultivated rice genomes**

| | Y476 | | | IRGC106162[9] | Yuanjiang wild rice[3] | T2T-Nip[21] | 9311[28] |
|---|---|---|---|---|---|---|---|
| | Hap1 | Hap2 | Primary | | | | |
| **Assembly** | | | | | | | |
| Genome size (Mb) | 411.1 | 411.9 | 418.8 | 377.1 | 376.5 | 381.7 | 391.8 |
| No. of chromosomes (gaps) | 12 (3) | 12 (3) | 12 (0) | * | 12 (1219) | 12 (0) | 12 (89) |
| Contig N50 (Mb) | 29.3 | 33.4 | 33.8 | 13.2 | 1.1 | * | 12.4 |
| Contig NG50 (Mb) | 29.3 | 33.4 | 36.3 | * | * | * | * |
| Genome BUSCO (%) | 98.7 | 98.5 | 98.8 | 96.2 | 97.36 | 98.8 | 98.7 |
| QV | 53.6 | 52.7 | 53.1 | * | * | 62.8 | * |
| LAI | 25.36 | 24.21 | 24.69 | 10.19 | 16.55 | 22.11 | 21.99 |
| Telomere number | 22 | 22 | 22 | 0 | 0 | 24 | 1 |
| Centromere number | 12 | 12 | 12 | 0 | 0 | 12 | * |
| **Annotation** | | | | | | | |
| No. of predicted genes | 36,150 | 36,336 | 36,422 | 33,903 | 34,830 | 55,986 | 41,319 |
| Average gene length (bp) | 2796 | 2793 | 2784 | 2397 | 2921 | 2941 | 3234 |
| Average CDS length (bp) | 1175 | 1173 | 1171 | * | 1125 | 1330 | 1063 |
| Protein BUSCO (%) | 97 | 97.1 | 98 | 93.4 | 94.2 | 92.0 | 95.7 |
| Repeat sequences (%) | 58.31 | 58.45 | 59.03 | 49.37 | 44.14 | 50.13 | 53.10 |

IRGC106162: an *O. rufipogon* accession collected from Laos by the International Rice Research Institute (IRRI); its genome was assembled by Xie et al.[9] at chromosome level. Yuanjiang wild rice: an *O. rufipogon* from Yuanjiang County, Yunnan Province, China; its genome was assembled by Li et al.[3] at chromosome level. *Data unavailable from the references.

wild rice components, 8.4% of *indica* components, 9% of *japonica* components, and 4% of *aus* components (Supplementary Fig. 1c).

## Sequencing, assembly, and annotation of a gap-free wild rice genome

Different sequencing platforms were applied to develop a high-quality genome assembly for Y476. Approximately, the Illumina data were used for *k*-mer analysis. The genome size of Y476 was estimated to be ~420 Mb with a heterozygosity of 0.86% (Supplementary Fig. 2). In all, 29.7 Gb (~70.9×) HiFi reads were generated by the PacBio sequel II platform (Supplementary Table 1), with 99.9% accuracy of long reads and an N50 length of 15,710 bp (Supplementary Fig. 3a, c). Nanopore ONT sequencing applying the latest ultra-long sequencing technology yielded an N50 length of 100,411 bp (Supplementary Fig. 3b, d), which was particularly suitable for the assembly of highly repetitive regions such as centromeric and telomeric regions, as well as the generation of a gap-free genome. For the Y476 gap-free genome assemblies, the preliminary assembly applied hifiasm with HiFi and ONT data, and generated contigs with an N50 length of 33.8 Mb, which was 3–30 times larger than that of reported wild rice genomes[3,9] (Table 1). To address the relatively high proportion of heterozygous fragments in the genome sequence, the assembled contig sequences were further filtered using "purge_dups". The chromosome ID and orientation were tuned in accordance with R498[16] by RagTag[17], and the final gap-free Y476 genome, comprised of 12 gap-free chromosomes, was generated. The de novo Bionano optical maps were aligned to the genome to verify the correct sequence and direction of the genome assemblies (Supplementary Fig. 4). Finally, the gap-free Y476 genome was assembled with a total length of 418.8 Mb.

Assembling haplotype-resolved genomes for highly heterozygous species can reveal a broad spectrum of variations and genes[18]. Hifiasm may be coupled with Hi-C reads to produce a pair of haplotype-resolved assemblies[19]. Hap1 and Hap2 yielded contigs with N50 lengths of 29.3 Mb and 33.4 Mb, respectively. Following the analysis pipeline described above, these contigs were assembled into 12 chromosomes, thereby giving rise to haplotype assemblies Hap1 (411.1 Mb) and Hap2 (411.9 Mb), which each had three gaps (Fig. 2 and Table 1). HiCPlotter[20] was applied to generate chromosomal interaction heatmaps of 12

chromosomes in two haplotype genomes, which showing no obvious mis-assembly (Fig. 2a, b).

We conducted transposable element (TE) and gene annotations on these three sets of genomes. First, we screened repetitive genome sequences to annotate TEs. The Hap1 genome comprised a total of 239,710,859 bp TEs, representing 58.31% of its total genome (Supplementary Table 2). Similarly, the Hap2 genome contained 240,727,453 bp TEs, accounting for 58.45% of its genome (Supplementary Table 2). Next, we applied a repetitive sequence mask and used the resulting sequences to predict gene structure. In terms of protein-coding genes, 36,150 were predicted in the Hap1 genome, featuring an average coding sequence size of 1175 bp and an average of 4.57 exons per gene (Supplementary Table 3). For the Hap2 genome, 36,336 protein-coding genes were predicted, with an average coding sequence size of 1173 bp and an average of 4.55 exons per gene (Supplementary Table 3). Furthermore, comparative genomic analysis highlighted 2,719,969 SNPs and 524,364 InDels in the Y476 Hap1 and T2T-Nipponbare[21] genomes, and 2,777,761 SNPs and 527,441 InDels were identified between the Y476 Hap2 and T2T-Nipponbare[21] genomes (Table 2). The same annotation process was applied to the gap-free primary Y476 genome, with the results detailed in the corresponding Supplementary Tables. The densities of GC pairs, genes, TEs, SNPs, and InDels in the Hap1 genome and Hap2 genome were plotted using 500 kb intervals across the 12 chromosomes (Fig. 2c, d).

## Quality assessment and validation of the Y476 assembly

The quality and completeness of the Y476 assembly were evaluated in multiple ways. We mapped HiFi, ONT, and WGS reads separately against the assemblies, yielding a mapping rate of over 99.13% and coverage rate (>1X) of over 99.96% for all three data types in both Hap1 and Hap2 genomes (Supplementary Table 4). Mapped reads demonstrated even coverage across both Hap1 and Hap2 genomes, with each of the three data types achieving nearly 90% coverage of the assembly (Supplementary Table 5).

The Hap1 and Hap2 had a QV[22,23] (quality value) of 53.6 and 52.7, LAI[24] (LTR Assembly Index) of 25.36 and 24.21 (Table 1), BUSCO[25] (Benchmarking Unique Single Copy Orthologs) score of 98.7% and 98.5% in genome mode (Supplementary Table 6), and 97.0% and 97.1%

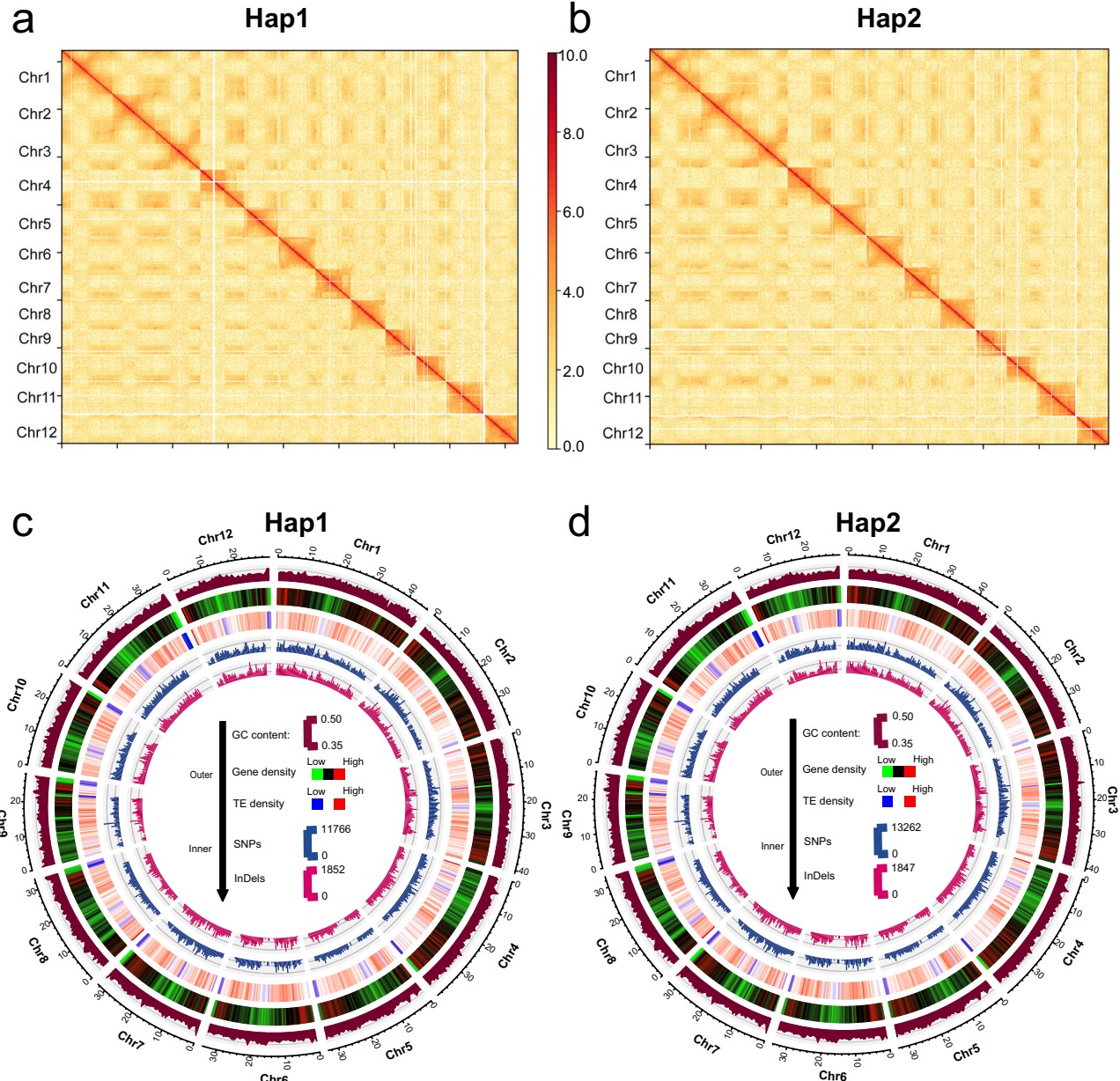

**Fig. 2 | Overview of the two haplotype genomes of Y476. a, b** Hi-C chromatin interaction map of the two haplotype genomes of Y476. **c, d** Circos plot of gene features at 500-kb intervals across the 12 chromosomes of the two haplotype genomes of Y476. The GC content, gene density, TE density, and SNPs and InDels between the Y476 and Nip genomes are shown (from the outer ring to the inner ring). The outer black track represents the chromosomes of the genome assembly (with units in Mb).

in the predicted gene, respectively (Table 1, Supplementary Table 7), demonstrating high accuracy and completeness of both assemblies.

We engaged the VerityMap[26] and T2T-Polish[22] pipelines to assess the quality of the genome, and the areas of errors identified in the results were incorporated into Supplementary Data 1. The VerityMap results reveal a combined length of possible heterozygous sites and errors amounting to 0.36 Mb (Hap1) and 0.15 Mb (Hap2), while the low-quality areas identified by T2T-Polish total 4.30 Mb (Hap1) and 4.18 Mb (Hap2). In addition, we identified the centromeric region on each chromosome (Supplementary Table 8). Using seven-base telomeric repeats (CCCATTT at the 5' end and TTTAGGG at the 3' end) as sequence queries, we identified 22 telomeres in the Y476 genome (Supplementary Table 9).

These results further demonstrated the high reliability and quality of the Y476 diploid genome assembly. We also evaluated two

published wild rice genomes[3,9] and gap-free primary Y476 genome using the same analysis, and the quality of Y476 was significantly better than any published wild rice genome[8,27] (Table 1).

## Global comparison and identification of genes and gene families from Y476

To dissect the genome variation between Y476 and cultivated rice, we compared the Y476 genome with *indica* (Xian) variety 9311[28] and *japonica* (Geng) variety Nip[21]. In comparison with 9311 and Nip, the Y476 genome contains more repeat sequences, resulting in a larger genome size. Synteny comparison of the genomic structure revealed high collinearity between Y476 and both Nip and 9311; fragment inversions, duplications and translocations are shown in Fig. 3a. Compared with Nip and 9311, Y476 has inversions on chromosome (Chr.) 6 and Chr. 9, and the inversion on chromosome 6 was verified by Hi-C and UL reads

**Table 2 | Global comparison of Y476 with Nip and 9311**

|  | Nip vs Hap1 | Nip vs Hap2 | 9311 vs Hap1 | 9311 vs Hap2 |
|---|---|---|---|---|
| No. of SNPs | 2,719,969 | 2,777,761 | 3,137,313 | 3,447,519 |
| No. of small InDels | 524,364 | 527,441 | 605,024 | 644,174 |
| Deletion number | 23,322 | 23,037 | 24,515 | 25,142 |
| Deletion total length (bp) | 90,158,394 | 89,406,408 | 101,500,659 | 102,526,535 |
| Deletion length range (bp) | 50–1,177,223 | 50–1,449,430 | 50–838,432 | 50–838,432 |
| Insertion number | 26,009 | 25,797 | 26,923 | 27,623 |
| Insertion total length (bp) | 118,400,368 | 119,533,919 | 121,014,120 | 122,515,479 |
| Insertion length range (bp) | 50–1,098,389 | 50–1,529,793 | 50–1,024,003 | 50–884,001 |
| Duplications | 560 | 544 | 483 | 461 |
| Inversions | 251 | 231 | 245 | 225 |
| Translocations | 301 | 285 | 379 | 280 |

(Supplementary Fig. 5). SNPs, small InDels and structural variations (SVs) are listed in Table 2. The similarity of the genomes of Y476 and Nip was greater than that between Y476 and 9311. SV analysis identified 49,331/48,834 SVs between Hap1/Hap2 and Nip, including 93/89 Mb of deletions and 122/120 Mb of insertions. Some randomly selected large SVs and some gap regions previously identified on the wild rice genomes were also verified by PCR amplification (Fig. 3b, Supplementary Fig. 6, and Supplementary Data 2). Moreover, 50,889 SVs were identified between Y476 and 9311[28] (Supplementary Fig. 7). Given the low completeness of the 9311 genome assembly, the identification of SVs may be biased, leading to a degree of inaccuracy.

We performed orthologous clustering of Y476, Nip, and 9311. In the Y476 genome, 27,454 gene families were identified, of which 20,675 families were common to all three genomes, whereas 303 families were specific to the Y476 genome (Supplementary Fig. 8). Among the gene families identified in the Y476 genome, 690 families were expanded and 247 families were contracted, including the NBS-LRR and RNA_pol_Rpc4 families, which play important roles in disease resistance[29] and grain regulation[30], respectively. An analysis of Gene Ontology (GO) terms for these specific families revealed that several biological processes, including DNA integration, viral genome integration into host DNA and ion transport, are enriched in the Y476 genome (Supplementary Table 10). A total of 5984 protein-coding genes absent from the Nip/9311 genomes were predicted in the Y476 genome. Of these genes, 18.1% (1085) were annotated by Pfam, and 6.9% (415) genes were annotated by GO terms. Importantly, 155 of 1085 annotated Y476 genes are potential disease resistance genes (R genes), including 106 NB-ARC genes and 49 genes containing an LRR domain, suggesting that these genes may confer the excellent disease resistance of Y476 (Supplementary Table 11).

We identified two large structural variations on Chr.4 and Chr.11, which contained tandem repeats of gene clusters. We found a gene cluster on Chr.4 (32 genes), including *LOC_Os04g32350*, which belongs to the RNA_pol_Rpc4 gene family and regulates the expression of genes involved in grain development[30]. The gene cluster on Chr.11 (39 genes) and three NBS-LRR genes present in Nip belong to a common gene family and were discovered by tandem repeat duplication of these three genes (Supplementary Table 12 and Supplementary Data 3). The number of NBS-LRR resistance genes in Y476 was significantly increased in comparison with Nip. By applying RNA-seq with rice blast challenge experiments, we revealed that the expression levels of these two gene clusters in Y476 were significantly different from their expression levels in Nip (Fig. 3c–f).

## Development of chromosome segment substitution lines in different cultivated rice genetic backgrounds

Based on the observation that there is wide genetic variation between wild rice accession Y476 and cultivated rice, it is anticipated that some

of these variations could be responsible for phenotypic differences, including resistance to biotic and abiotic stresses. In order to dissect wild rice genomic information and manipulate these variations for directional breeding, we developed two sets of chromosome segment substitution lines (CSSLs), using Y476 as the donor parent and 9311 and Nip as the receipt/recurrent parent, respectively. The CSSL generation procedure is summarized in Supplementary Fig. 9. The CSSL/9311 population is an advanced backcrossed generation population that contains 198 lines and covers 85% of the wild rice genome; each line has an average of 96.5% recurrent parent genome and harbors 7.3 substitution segments (Fig. 4a–c and Supplementary Tables 13 and 14). The CSSL/Nip population, a less backcrossed population, contains 225 lines and covers the whole genome of wild rice; each line has an average of 80.6% recurrent parent Nip genome and harbors 21 substitution segments (Fig. 4d). The genetic constitution of the CSSL/Nip population was analyzed. In the CSSL/Nip population, the sizes of the substituted segments ranged from 0.37 to 106 cM, with an average of 14.4 cM. Twenty-six percent of substituted segments were smaller than 5 cM, 22% of substituted segments ranged from 5 to 10 cM, and 11% were larger than 30 cM. 40% of CSSLs contained no more than 20 substituted segments (Fig. 4d–f and Supplementary Tables 15 and 16).

QTL mapping was performed based on phenotype and genotype association using the two CSSL populations described above. Some genes associated with domestication-related traits, such as *sd1* for plant height[31,32], *sh4* for seed shattering[33], and *C1* for red or purple coloration[34,35], were observed in the CSSL populations and fine-mapped to their exact positions (Supplementary Fig. 10). Variants of these three loci from the CSSLs have the wild-type haplotypes which were previously reported. These results demonstrate that the CSSLs generated in this study were useful resources for the identification of wild rice genes.

## QTL mapping of favorable agronomic traits using CSSLs

Using the CSSL/Nip population, nine agronomical traits, including grain length, grain width, 1000-grain weight, grain length to grain width ratio, plant height, panicle length, tiller number, length of flag leaf and width of flag leaf were investigated in three environments and showed a large range of variation (Supplementary Fig. 11 and Supplementary Table 17). We identified 244 QTLs associated with these nine agronomic traits in three environments (Fig. 5a, Supplementary Fig. 12, and Supplementary Data 4). Among these QTLs, 223 were previously uncharacterized. Approximately 130 genes that were not found in the cultivated rice genomes were identified in these QTLs, and these loci from wild rice provide a useful resource for further studies. SV identification on the QTL intervals revealed 1323 SVs distributed on 199 QTLs on 12 chromosomes (Fig. 5b and Supplementary Data 5).

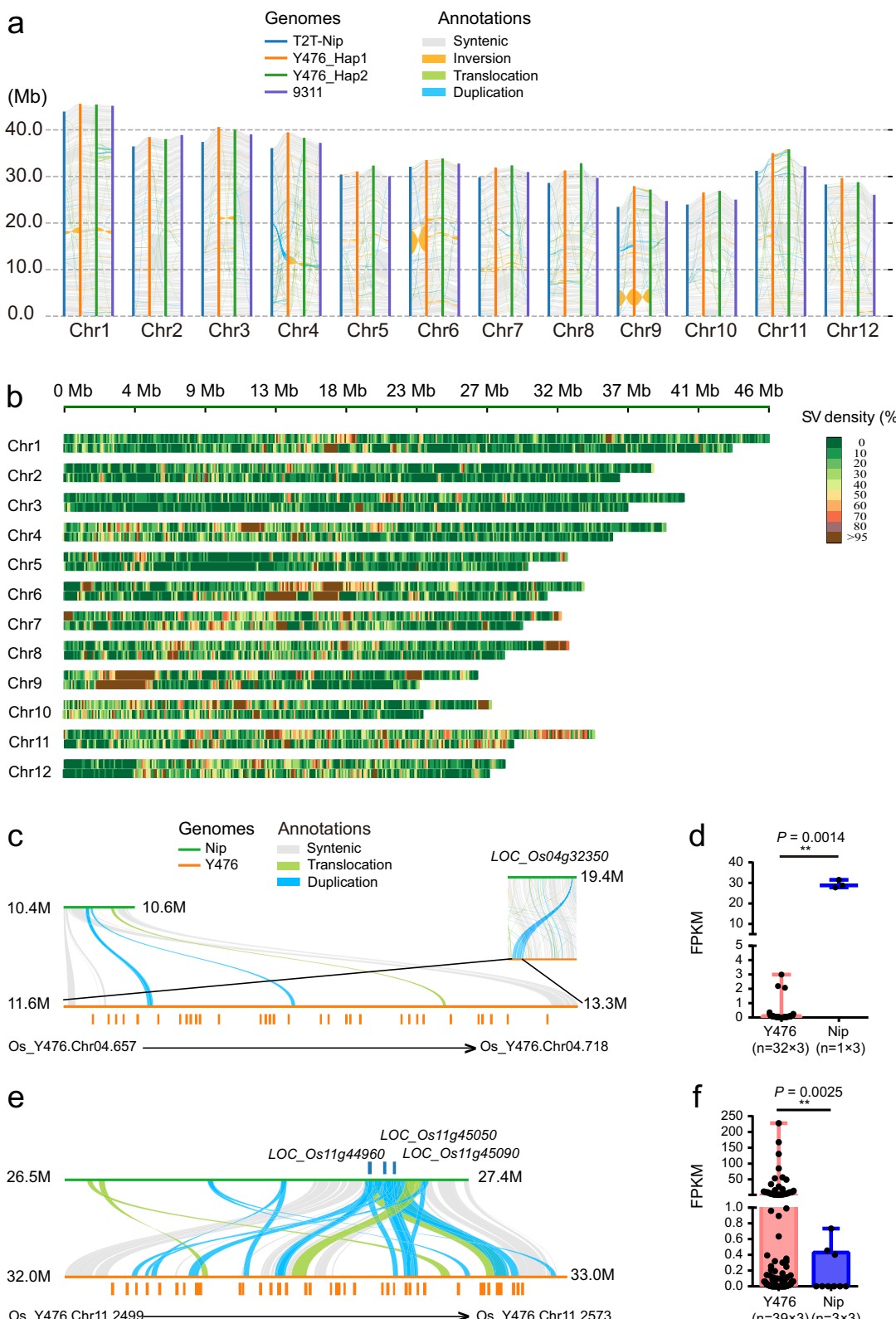

**Fig. 3 | Characterization of genomic variations between *O. rufipogon* Y476 and *O. sativa* (*Xian* 9311 and *Geng* Nip).** a Syntenic blocks shared between Y476 and 9311/Nip. Gray lines connect matched gene pairs. Inversion blocks are highlighted in orange. The translocation and duplication blocks are highlighted in light green and blue, respectively. **b** SVs between Y476 and Nip for each chromosome. The heatmap above shows the SV density in Y476, and the heatmap below shows the SV density in Nip. **c** Identification of a tandem repeat gene cluster related to rice grain regulation on Chr.4 of Y476. **d** Expression analysis of tandem repeat gene clusters related to grain regulation in Y476 and the corresponding homologous genes in

Nip. "*n*" represents the number of genes in the tandem repeat gene cluster multiplied by replicates. Comparisons were performed by two-tailed Student's *t* test (*P < 0.05, **P < 0.01). **e** Identification of tandem repeat gene clusters related to disease resistance on Chr.11 of Y476. **f** Expression analysis of tandem repeat genes with LRR domains in Y476 and the corresponding homologous genes in Nip after inoculation with *M. oryzae*. "*n*" represents the number of genes in the tandem repeat gene cluster multiplied by replicates. Comparisons were performed by two-tailed Student's *t* test (*P < 0.05, **P < 0.01).

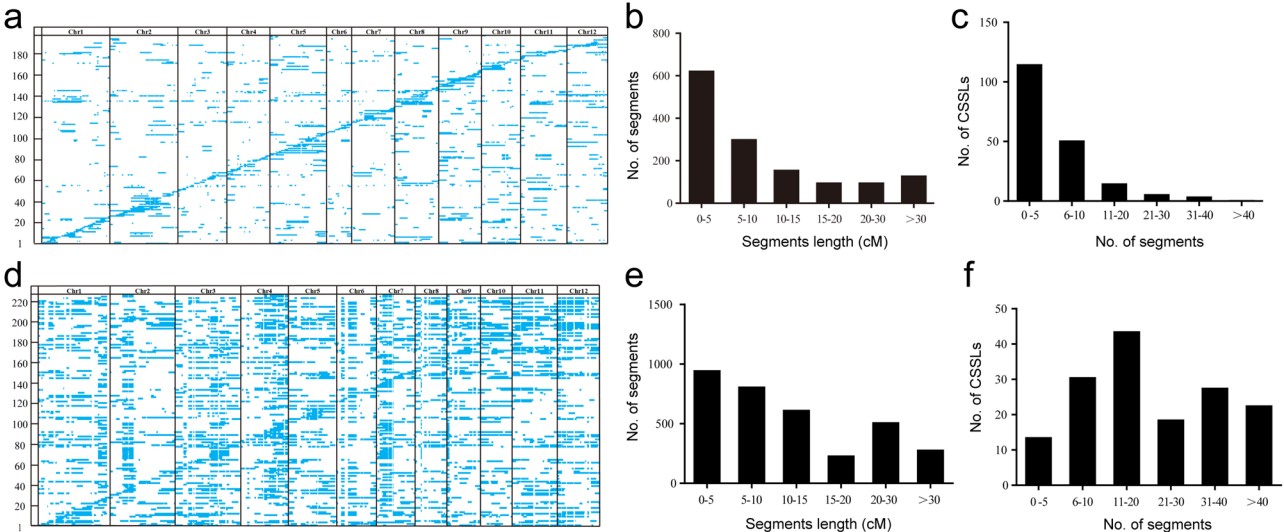

**Fig. 4 | Genotypes and frequency of the substituted segments of two sets of CSSLs. a** Genotypes of the CSSL/9311 population. Regions with a white background represent homozygous segments from 9311, and blue regions indicate homozygous segments from Y476. The horizontal axis indicates one CSSL, and the vertical axis indicates one substituted segment of wild rice. **b, c** Frequencies of the substituted chromosome segments of the CSSL/9311 population according to genetic length (**b**) and segment number (**c**). **d** Genotypes of the CSSL/Nip population. Regions with a white background represent homozygous segments from Nip, and blue regions indicate homozygous segments from Y476. The horizontal axis indicates one CSSL, and the vertical axis indicates one substituted segment of wild rice. **e, f** Frequency of substituted chromosome segments of the CSSL/Nip population according to genetic length (**e**) and segment number (**f**). Source data are provided as a Source Data file.

The SVs for several documented genes in QTLs between Y476 and Nip were confirmed. Analysis of RNA-seq data from Nip and Y476 demonstrated that more than half of the differentially expressed genes (DEGs) had at least one SV, and 82% of the DEGs harbored SVs in their promoter regions, suggesting that SVs play important roles in regulating the expression of these genes. For example, a known *OsSWEET14* gene (*LOC_Os11g31190*), which encodes a sugar transporter and was shown to negatively regulate grain weight[36], harbors a 663 bp SV upstream. A CSSL with this allele from Y476 showed reduced expression of the encoded gene (Supplementary Fig. 13). Therefore, this SV may be the cause of the significant difference in the expression of the *OsSWEET14* gene in Y476 in comparison with Nip.

## Identification of chromosome loci controlling salt tolerance using CSSLs

To identify beneficial alleles for abiotic stress tolerance, salt tolerance was investigated under 85 mM NaCl (0.5% salt stress) during the entire growth period using the CSSL/Nip population. Phenotypic transgressive variation was observed in the CSSL/Nip population, in which nearly half of the CSSLs were more salt tolerant than their recurrent parent Nip, while the salt tolerance of approximately half of the CSSLs was similar to that of Nip (Supplementary Fig. 14a). Three QTLs were identified, and one QTL near locus *S2_4579633*, with the highest LOD value (12.3), was selected (Fig. 6a and Supplementary Table 18). Four genes were found in this interval (Supplementary Table 19), and the variations in these genes between wild rice and Nip were investigated. One CSSL line, N133, which harbors this locus and was highly salt tolerant (Level 3), was selected for further study (Supplementary Fig. 14b, c). The salt tolerance of 20-day-old N133 seedlings was higher than that of Nip, and the survival rate, fresh weight and dry weight of above-ground N133 seedlings were much higher than those of Nip under salt stress (Fig. 6b–f).

Based on the transcriptomic data, we determined the relative expression levels of four genes in the selected QTL near locus *S2_4579633*, but only one gene, *LOC_Os02g08540*, had a significantly different transcript level between N133 and Nip under the salt treatment (Supplementary Fig. 14d–g). Phylogenetic analysis showed that *LOC_Os02g08540* encodes an unknown expressed protein that has not been reported in the *Oryza* family or other species (Supplementary Fig. 15). Real-time PCR assays confirmed that *LOC_Os02g08540* expression was strongly induced by salt stress in N133 plants (Fig. 6g). Analysis of the genomic sequences of Y476 and Nip revealed that an 87-bp deletion and a 240-bp deletion were present in the promoter region of *LOC_Os02g08540* and its downstream region (Fig. 6h), respectively. Furthermore, the 87-bp SV was present in both haplotype genomes of Y476. At the same time, the promoter and CDS region of *LOC_Os02g08540* in Y476 were amplified and sequenced, the results were consistent with Y476 genome assembly, and compared with Nip, there are three non-synonymous mutations in its CDS region (Supplementary Table 20). Therefore, integrating the transcriptome and genome variation data identified *LOC_Os02g08540* as a candidate gene associated with salt tolerance.

Line N133 showed excellent salt tolerance at the seedling and adult plant stages in the field, and it survived under 85 mM NaCl (0.5% salt stress) during the entire growth period (Supplementary Fig. 14b, c). In addition, the yield traits of line N133 were also better than those of Nip under normal conditions, and the yield per plant of N133 was significantly higher than that of Nip (Supplementary Fig. 16). Next, based on further analysis of the CSSL/9311 population, line C58 was selected as a near-isogenic line (NIL) of wild rice *LOC_Os02g08540*. The expression level of *LOC_Os02g08540* in C58 was significantly higher than that in 9311 before and after salt treatment. In addition, C58 exhibited better salt tolerance than 9311, but the agronomic traits of these lines did not differ significantly (Supplementary Fig. 17). These results provide promising gene and germplasm resources for future breeding programs aimed at producing rice varieties with improved salt tolerance.

## Identification of rice blast-resistant genes from wild rice using CSSLs

We identified seven QTLs for rice blast resistance, which contained 70 open reading frames. The QTL located near *S7_21365207* has the largest LOD value (10.9) and the greatest proportion of phenotypic variance explained (PVE) (12.1%) (Fig. 7a and Supplementary Table 21). CSSL/Nip

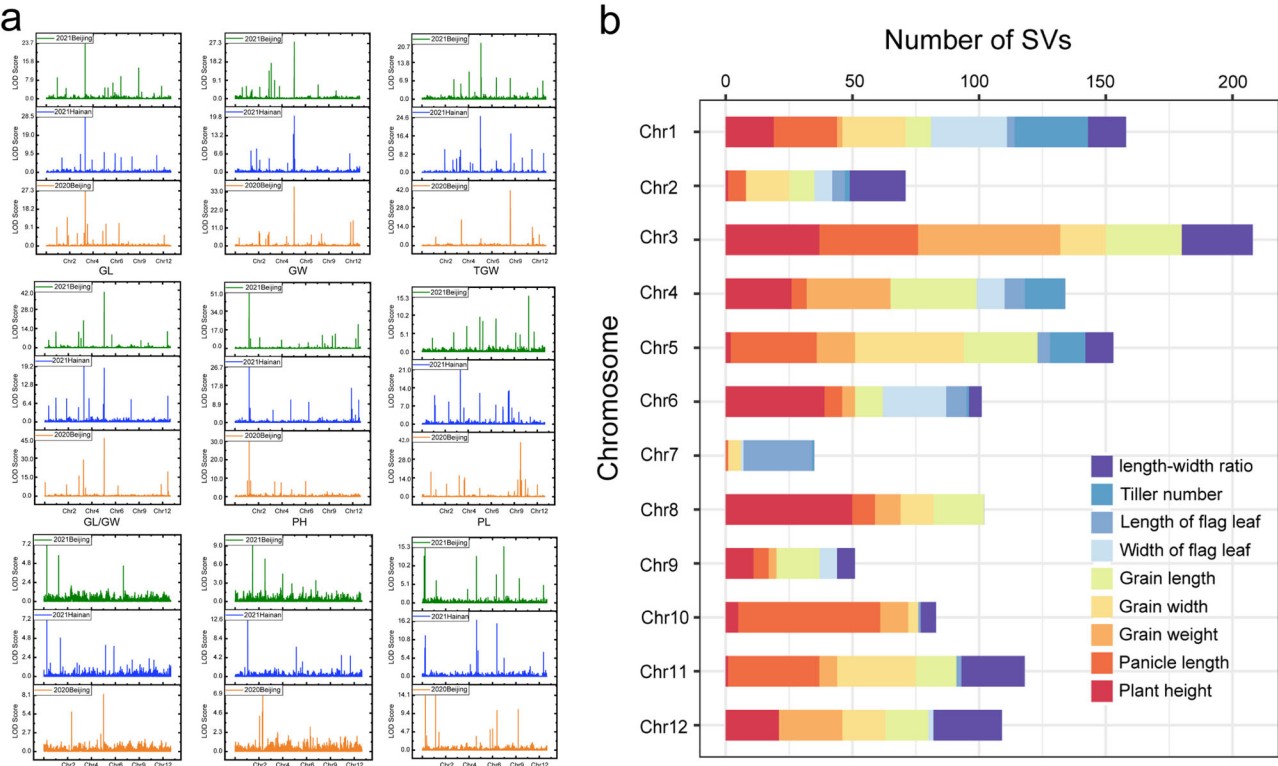

**Fig. 5 | Identification and analysis of QTLs related to agronomic traits. a** QTL mapping for grain length (GL), grain width (GW), 1000-grain weight (TGW), length-width ratio (GL/GW), plant height (PH), panicle length (PL), tiller number (TN), length of flag leaf (LFL), and width of flag leaf (WFL) in three environments. The *x* axes show the introgression segments on 12 chromosomes and the *y* axes show the logarithm of odds (LOD) score. **b** SV number analysis on 12 chromosomes for nine agronomic trait-related QTLs. Source data are provided as a Source Data file.

line N154, which harbors this QTL and was highly resistant to rice blast, was selected for further study. The lesion length online N154 was very close to zero (Fig. 7b). Three genes were determined to be located near the *S7_21365207* locus: *LOC_Os07g35660*, *LOC_Os07g35670* and *LOC_Os07g35680* (Supplementary Table 22). Only *LOC_Os07g35680* had a significant difference in expression level between N154 and Nip, and this difference was confirmed by real-time PCR (Fig. 7c and Supplementary Fig. 18). Subsequently, the CDS region and promoter region of *LOC_Os07g35680* of Y476 were amplified and sequenced, and the results were consistent with the assembly sequence of Y476 genome. Furthermore, a 7.8-kb SV was found in the first intron of *LOC_Os07g35680* between N154/Y476 and Nip (Fig. 7d), and the SV is present in the Hap2 genome of Y476 but is absent in Hap1. In addition, variants in the CDS region between N154 and Nip are listed in Supplementary Data 6. Real-time PCR analysis revealed that the expression level of *LOC_Os07g35680* in the leaves of N154 was significantly higher than that in Nip leaves, and *LOC_Os07g35680* expression was significantly increased after rice blast treatment (Fig. 7c, e). Based on these results and the genome variation data, *LOC_Os07g35680* was identified as a candidate gene from wild rice for rice blast resistance. *LOC_Os07g35680* encodes a receptor-like kinase (RLK). Phylogenetic analysis revealed that *LOC_Os07g35680* belongs to the RLK family and is clustered with several RLK genes in the rice genome (Supplementary Fig. 19). To confirm the function of *LOC_Os07g35680*, a NIL (9311 background) for the wild rice *LOC_Os07g35680* gene was developed from the CSSL/9311 population. The resistance of NIL-*LOC_Os07g35680* to rice blast was stronger than that of 9311. In addition, the expression level of *LOC_Os07g35680* in the leaves of NIL-*LOC_Os07g35680* was consistently higher than that of 9311 before and after *M. oryzae* inoculation (Supplementary Fig. 20).

We next compared the transcriptomes of N154 and Nip after rice blast infection using RNA-seq data. Global gene expression differences were found between Nip and N154. The analysis revealed 3788 DEGs in N154 without blast infection in comparison with Nip. Among this set of genes, 841 DEGs were identified in N154 following blast infection in comparison with Nip, including 87 DEGs enriched in the GO terms "cell death", "response to stimulus", "defense response" and "immune response" (Fig. 7f and Supplementary Fig. 21a, b). The set of 841 DEGs contained four cloned genes related to rice blast resistance; *Pi9*[37] was significantly up-regulated in N154, while *OsWAK112d*[38], *OsMADS26*[39,40] and *Pish*[41,42] were significantly down-regulated (Supplementary Fig. 21c). The expression patterns of these four rice blast resistance genes were confirmed by real-time PCR (Fig. 7g).

Using CRISPR/Cas9 technology, we generated knock-out mutants of *LOC_Os07g35680* in the N154 background (N154-KO) (Supplementary Fig. 22). As expected, the rice blast resistance of the knock-out plants decreased significantly. The N154-KO lines displayed enhanced susceptibility to *M. oryzae* similar to Nip, and the lesion length on the N154-KO lines was significantly increased compared with that of N154 (Fig. 7h, i). Subsequently, we detected the expression levels of these four rice blast resistance genes in the N154-KO lines. Only *OsMADS26* expression was significantly changed in the N154-KO lines, and the change was consistent with that of Nip, which was significantly different from that of N154 (Fig. 7j). These results demonstrate that the natural variation in the rice *LOC_Os07g35680* locus is critical for resistance to *M. oryzae* and leads to changes in the transcription of rice blast resistance gene *OsMADS26*, which might contribute to the superior disease resistance in N154 and wild rice.

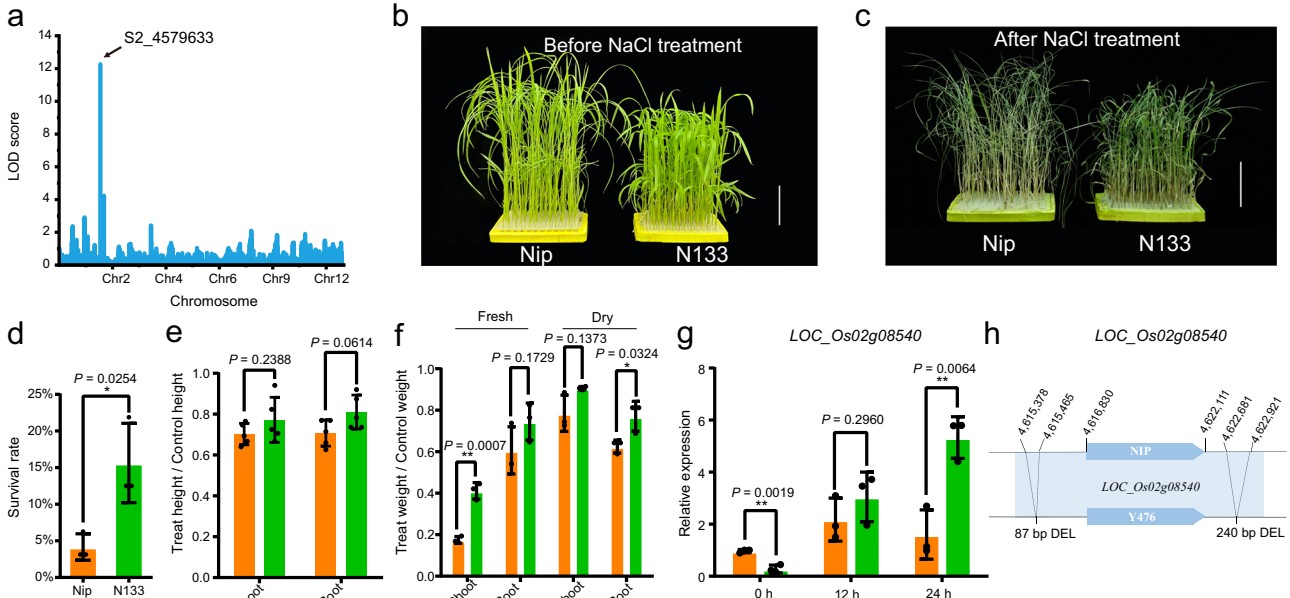

**Fig. 6 | Identification of wild rice salt tolerance genes. a** QTL mapping for salt tolerance. The *x* axis shown the introgression segments on 12 chromosomes and the *y* axis show the logarithm of odds (LOD) score. **b**–**f** N133 exhibited significantly higher salt tolerance in comparison with Nip. Twenty-day-old seedlings of the Nip and N133 lines were treated with 150 mM NaCl for five days (**b**) and recovered in fresh water for 5 days (**c**). The survival rate (**d**), relative plant height and root length (**e**), and fresh and dry weight (**f**) were determined. The scale bars correspond to 5 cm in (**b**, **c**). The results are presented as the mean ± SD from three biological replicates (*n* = 3) in (**d**, **f**). The results are presented as the mean ± SD from five biological replicates (*n* = 5) in (**e**). All comparisons were performed by two-tailed Student's *t* test (\**P* < 0.05, \*\**P* < 0.01). **g** Expression levels of *LOC_Os02g08540* in Nip and N133 at 0 h, 12 h, and 24 h after treatment with 150 mM NaCl. The results are presented as the mean ± SD from three biological replicates (*n* = 3). Comparisons were performed by two-tailed Student's *t* test (\**P* < 0.05, \*\**P* < 0.01). **h** Sequence variation of *LOC_Os02g08540* in Nip and Y476/N133. Source data are provided as a Source Data file.

## Discussion

Asian cultivated rice *O. sativa* was domesticated from wild rice *O. rufipogon*, and this process likely occurred in the global center of wild rice diversity in southern China. The genomes of wild rice strains harbor abundant beneficial alleles that have been lost during the breeding of modern varieties, which represent a useful resource for breeding programs aimed at producing strains with better resistance to biotic and abiotic stressors, as well as for research aimed at understanding the mechanisms underlying stress resistance in plants. Therefore, the primary objectives of this study were to construct a platform and genomic and germplasm resources for high-throughput gene identification from *O. rufipogon*. The chosen strategy for this study involved a comprehensive analysis pipeline that progressed from the construction of a high-quality reference genome to a comparative analysis of the transcriptomes of distinct wild and domesticated populations under stress, followed by QTL identification, gene annotation and the identification of likely mechanisms underlying enhanced stress resistance.

A high-quality reference genome is critical for studies aimed at understanding genome structure and genetic variation. However, assembling a typical *O. rufipogon* genome has been challenging because of its high degree of heterozygosity. In the past five years, well-documented genome assemblies of *O. rufipogon* accessions have been produced, including W1943 with a contig N50 of 34 kb[5,7], an accession from Yunnan province, China, with a contig N50 of 1.1 Mb[3], and IRGC106162 from Laos, with a contig N50 of 13.2 Mb[9]. However, none of these accessions represent typical Chinese wild rice, and a gap-free assembly has not been produced.

In this study, we chose wild rice accession Y476, a typical Chinese wild rice strain with high resistance to biotic and abiotic stresses, for genome assembly (Fig. 1). We explored which subpopulation Y476 belongs to based on Huang's 446 wild rice (*O. rufipogon*) sequencing data and the Or-I, Or-II, and Or-III classification[2], but the results seem to

be less clear-cut. Meanwhile, we also noticed that the relatively low sequencing quality of 446 wild rice samples may have led to a certain degree of bias and inaccuracy in the subgroup status analysis of Y476. Therefore, an accurate definition of Y476 belonging to the subpopulation may require a larger wild rice population and higher-quality sequencing data. Notably, the Y476 genome can serve as an important reference genome for the next step of wild rice classification and data analysis. The Y476 plant used for genome assembly in this study had a relatively high degree of heterozygosity (0.86%). We de novo assembled its haplotype-resolved gapless genome (contigs with an N50 length of 33.8 Mb) by combining the latest methods for obtaining HiFi reads, ONT ultra-long sequencing, and genome assembly. The genome of Y476 featured evident improvements in continuity and quality compared with existing wild rice genomes[3,9] (Fig. 2 and Table 1) and showed a high degree of synteny with those of *japonica* and *indica*. One important application of high-quality de novo assembly of a reference genome is the detection and characterization of genetic variations in the whole genome, especially SVs. Recent studies have shown that the impacts of SVs on genomic polymorphisms and functional gene variation are greater than those of SNPs[43]. SVs have been found to affect many rice agronomic traits, including hybrid sterility, flowering time, grain size, and disease resistance[44,45]. Our global comparison of the genomes of Y476 with *japonica* and *indica* revealed abundant SVs, genes and gene families that were absent in the cultivated rice genomes (Fig. 3). These results suggest that many wild rice genes were lost or changed during rice domestication, and many of these genes may be exploited to improve modern rice varieties.

The development of CSSLs is laborious. The two CSSL populations we constructed over the last ten years provide an experimental platform and valuable resources for the dissection of wild rice genomes and the identification of elite alleles. CSSLs that were relatively less backcrossed and covered the whole wild rice genome were used for QTL/gene identification, and the advanced backcrossed CSSLs

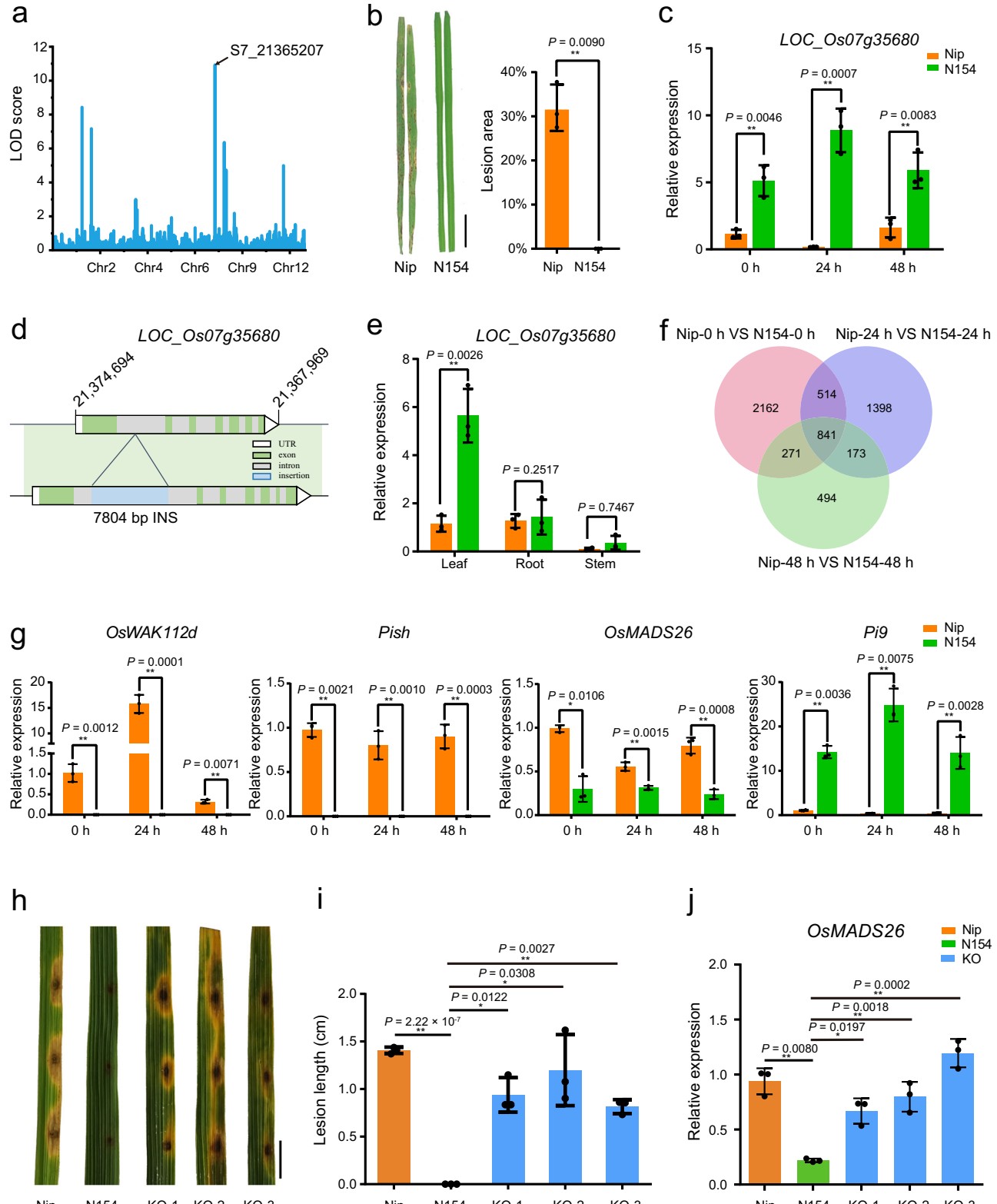

provided single-segment substitution lines or near-isogenic lines of wild rice genes (Fig. 4). Many key genes involved in domestication were easily identified using these two CSSL populations, demonstrating that CSSLs are effective for gene mapping. QTL identification was performed using the CSSL population, and abundant SVs were found in the QTLs according to the genomic data. Combining the genomic data and CSSLs allowed us to identify QTLs/genes associated with agronomic and yield traits and salt stress, and also to show that SVs play an

important role in gene expression regulation (Figs. 5 and 6). Moreover, we identified *LOC_Os07g35680*, a rice blast resistance gene. The wild rice *LOC_Os07g35680* allele has a 7.8-kb insertion in its intron region, which may be the cause of its elevated expression level. Further study showed that *LOC_Os07g35680* improved rice blast resistance, and this effect might be mediated by inhibition of the expression of *OsMADS26*, a negative regulatory factor of rice blast disease (Fig. 7). These QTLs and genes provide a framework for detailed functional analysis of

**Fig. 7 | Identification and function analysis of rice blast resistance genes. a** QTL mapping for rice blast resistance. The *x* axis shows the mapping on 12 chromosomes. The *y* axis shows the logarithm of odds (LOD) score. **b** Disease symptoms and lesion area of Nip and N154 after inoculation with *M. oryzae* isolates FJ07-5-2 and FJ07-8-1. Lesion area was measured at 7 dpi. The results are presented as the mean ± SD from three biological replicates (*n* = 3). Comparisons were performed using two-tailed Student's *t* test (*\*P* < 0.05, *\*\*P* < 0.01). **c** Expression levels of *LOC_Os07g35680* in Nip and N154 at 0 h, 24 h, and 48 h after rice blast inoculation. The results are presented as the mean ± SD from three biological replicates (*n* = 3). Comparisons were performed using two-tailed Student's *t* test (*\*P* < 0.05, *\*\*P* < 0.01). **d** Sequence variation of *LOC_Os07g35680* in Nip and Y476. **e** Expression patterns of *LOC_Os07g35680* in different tissues of Nip and N154. The results are presented as the mean ± SD from three biological replicates (*n* = 3). Comparisons were performed using two-tailed Student's *t* test (*\*P* < 0.05, *\*\*P* < 0.01). **f** Venn diagram showing the overlap of DEGs from the transcriptomic data of Nip and N154 at 0 h, 24 h, and 48 h after rice blast inoculation. Two-week-old rice leaves were used for transcriptome assays. **g** Expression levels of four rice blast resistance-related genes in Nip and N154 at 0 h, 24 h, and 48 h after rice blast inoculation. The results are presented as the mean ± SD from three biological replicates (*n* = 3). Comparisons were performed using two-tailed Student's *t* test (*\*P* < 0.05, *\*\*P* < 0.01). **h, i** Punch inoculation of Nip, N154 and *LOC_Os07g35680* knock-out plants (**h**). Lesion length was measured at 6 dpi. **i** Blast isolates FJ07-5-2 and FJ07-8-1 were used for inoculation. Scale bar correspond to 1 cm. The results are presented as the mean ± SD from three biological replicates (*n* = 3). Comparisons were performed using two-tailed Student's *t* test (*\*P* < 0.05, *\*\*P* < 0.01). **j** Expression levels of *OsMADS26* in Nip, N154 and N154-KO lines. The results are presented as the mean ± SD from three biological replicates (*n* = 3). Comparisons were performed using two-tailed Student's *t* test (*\*P* < 0.05, *\*\*P* < 0.01). Source data are provided as a Source Data file.

genomic segments in *O. rufipogon* and should be exploited further for future rice breeding.

In summary, we assembled a haplotype-resolved gapless genome of a typical Chinese common wild rice strain, which we used to identify extensive variations between wild and cultivated rice. These variations were introgressed into two cultivated rice genetic backgrounds by constructing CSSL populations. Our results highlight the role of SVs in functional gene variation. We explored beneficial genomic sequences conferring resistance to biotic and abiotic stresses, which could be used in breeding programs to produce plants with desirable traits. The reference genome and CSSL populations published herein will accelerate wild rice functional genomics studies and genome-enabled improvement of stress resistance.

## Methods

### Plant materials, DNA extraction, and library construction

The *O. rufipogon* accession Y476 was originally collected in Sanya, Hainan Province, China (N18.15°, E109.31°), and conserved in the Chinese National Wild Rice Germplasm Garden. For genome sequencing, high-quality genomic DNA was extracted from leaves using a modified CTAB method[46]. The quality of DNA was checked by agarose gel electrophoresis, and DNA was sequenced with both Illumina HiSeq X Ten (Illumina Inc., San Diego, CA) and PacBio Sequel (Pacific Biosciences of California, Menlo Park, CA) platforms. A portion of the DNA was sent to Frasergen to construct circular consensus sequencing (CCS) libraries and sequence them using a PacBio Sequel platform and Illumina Hiseq platforms, and another portion of DNA was sent to Benagen to construct libraries and sequence them using ONT Ultra-long reads. Short reads generated from the Illumina platform were to estimate the genome size, level of heterozygosity, mapping rate, and coverage. Long reads from the PacBio platform were used for genome assembly. Trizol (Invitrogen) was used to extract RNA from rice at the seedling, tillering, and heading stages. Panicles, leaves, stems, and roots were collected for RNA sequencing. The RNA-seq libraries were prepared with an insertion size of 300 bp and sequenced on the Illumina platform.

### Genome estimation using *k*-mer analysis

Illumina short reads were used for the survey analysis. The *k*-mer distributions were estimated using Jellyfish[47] with parameters -m 21 -t 1 -s 5 G -C. The genome size and heterozygosity were calculated by GenomeScope with default parameters[48].

### Genome de novo assembly and quality assessment

For the Y476 diploid haplotype-resolved genome assemblies, we generated HiFi reads from PacBio Sequel II system, ultra-long reads from Nanopore sequencing, and Hi-C sequencing. The long (>15 kb) and highly accurate (>99%) HiFi reads, ultra-long reads (>100 kb), and Hi-C reads were assembled with Hifiasm using parameters −h1 −h2 −ul,

producing one primary genome and two haplotype draft contig genomes[19]. Hi-C data were parsed into valid and invalid interaction pairs using Hi-C-Pro v2.11.1[49], retaining only the valid pairs for further assembly. Some contigs were discarded due to their short (<200 kb) and redundancy (high similarity). This resulted in Primary, Hap1, and Hap2 contigs genomes. The primary assembly chromosome ID and orientation were adjusted in alignment with the R498[16] rice genome by RagTag[17]. 3D-DNA[50] was used to connect and order contigs (from longest to shortest), forming pseudomolecules for the Hap1 and Hap2 genomes. The heatmap of genomic interactions was plotted by HiC-Plotter software[20]. The Primary genome and haplotype-resolved genome were polished using short reads, and HiFi reads by Racon and Merfin with two iterative rounds[22].

Genome completeness was evaluated by BUSCO using the "embryophyta_odb10" database[25]. Genome continuity was evaluated by calculating the contig N50 length. The accuracy of the genome was evaluated by mapping the WGS sequencing data to the genome and calculating the mapping rate and coverage by qualimap2[51,52]. Finally, the LAI value method was used to evaluate the assembly level of the genome based on repeat sequences[24]. We also assessed the genome assembly using Merqury QV based on the 21-mer hybrid Merqury *k*-mer database by combining Illumina PCR-free and HiFi reads[22,23]. The VerityMap[26] and the T2T-Polish[22] pipelines, which rely on long-read data, were employed to verify the correspondence between the assembly and the long reads.

### Identification of centromere and telomere sequences

We obtained the 155–165 bp CentO satellite DNA sequences in rice, and HMMER was used to search for the locations of centromeres in our reference genome[53]. Centromeric regions were defined as regions containing all units with high hit scores. The telomeric sequence 5′-CCCTAAA-3′ and the reverse complement of these seven bases were searched directly.

### Genome annotation

RepeatMasker was used to mask the genome and annotate TE elements based on a rice high-quality non-redundant TE library[54]. This TE library was generated by EDTA, which has been validated using the MSU rice genome and was found to be robust for both plant and animal species[55].

We predicted the structures of protein-coding genes in the rice genome using three gene prediction methods: ab initio prediction, homology-based prediction, and RNA-seq analysis. Before gene prediction, the assembled genome was hard and soft-masked using RepeatMasker[54]. We adopted Augustus (unsupervised training), GlimmerHMM, and Braker2 (prediction based on transcriptome data) to perform ab initio gene prediction[56–58]. Exonerate[59] (v2.2.0) was used to conduct homology-based gene prediction. The protein sequences from the MSUv7, MH63RS2, ZS97RS2, and R498 rice genomes, as well

as the TAIR10 *Arabidopsis thaliana* genome, were aligned to our genome assembly, and coding genes were predicted using Exonerate with default parameters[59]. The sets of RNA sequencing data were each approximately ~6 Gb in size. The transcriptome pipeline included de novo assembly of transcripts (based on Trinity to PASA) and genome-guided assembly (Hisat2 to StringTie to Transdecoder)[60–63]. EVidenceModeler was used to integrate all prediction results from the three methods to predict gene models[64]. Finally, gene models were filtered by removing gene coding sequences that overlapped with TE sequences by more than 20%, as well as those with a coding region that was shorter than 150 bp.

Three methods were used to predict the functions of protein-coding genes. First, BlastP was used to search the sequences against protein sequences in the NCBI non-redundant protein database (NR) and Swiss-Prot (http://web.expasy.org/docs/swiss-prot/guideline.html) database[65]. Second, protein domain and gene ontology term annotations were performed using InterProScan[66]. Third, KEGG annotation with the KEGG Automatic Annotation Server was used to identify significantly enriched cell signaling pathways among sets of genes[67]. tRNAscan-SE was used to identify tRNA genes with default parameters[68]. RNAmmer was used to predict rRNA sequences[69].

## Synteny analysis

Nucmer was used to perform comparisons between genomes with parameters –mum –mincluster 200 –minmatch 100[70]. Delta-filter was used to filter the results of the alignment file produced by nucmer with parameters -i 95 -l 100 -1. The dotplot was generated by the function mummerplot in MUMmer4[70].

## Genome-wide comparisons and identification of SNPs, InDels, and structure variations

MUMmer4[70] was used to compare the genomes of Y476 and Nip with parameters -maxmatch -c 100 -l 50. Next, we filtered the delta files using the delta-filter -1 parameter, which kept only the best alignments. Finally, we identified SNPs and InDels from the filtered delta files using "show-snps" with the "-ClrT" parameter. Structure variations were identified using MUMandCo[71] with default parameters based on the whole-genome alignment information provided by MUMmer (v4)[70].

## Distribution of structure variations relative to gene position and annotation

We classified SVs into five categories based on their overlapping genomic regions (coding region, intron, ± 2 kb of genes, and intergenic regions). If a SV shared overlaps with two or more different genomic regions, the SV was classified into different genomic regions at the same time. The percentage of each SV category was calculated. If a SV overlapped with the gene region or the upstream or downstream 2 kb region, we classified it as an SV with an effect on the gene of interest, and we annotated the SV with that gene.

## SV validation

To align the HiFi and ONT data of Y476 with the Y476 and Nip genomes, minimap2[72] was employed by utilizing the parameters -ax map-hifi and -ax map-ont, respectively. Subsequently, the aligned data were visualized using the Integrative Genomics Viewer (IGV)[73]. In addition, Y476's Hi-C data were aligned to the Y476 and Nip genomes using Hi-C-Pro[49], and the interaction matrices were visualized using HiCPlotter[20].

The two ends of SV were amplified by PCR, and the primers were designed based on the two ends of insertion sequence and two genomic common regions adjacent to the insertion ends. Then the DNA of Nip and Y476 were used as templates for amplification, and the amplified products were subsequently detected by agarose gel electrophoresis.

## Construction and genotyping of CSSLs

The CSSL populations used in this study were constructed with wild rice Y476 as the donor parent, and 9311 and Nip as the respective recipient parents. A schematic of the development of CSSLs is shown in Supplementary Fig. 9. A whole-genome survey was performed using SSR and InDel molecular markers during the construction of the CSSL/9311 population. A whole-genome survey of the CSSL/Nip population was performed by genome sequencing. We eliminated lines with a high level of genetic background noise and selected lines with relatively few segments, which were self-pollinated to produce homozygous lines as much as possible.

Genotype identification of the CSSL/Nip population was completed by genotyping by target sequencing (GBTS) with 10k SNPs. To identify introgression segments from Y476 to Nip, a sliding-window approach was applied[74]. Based on the SNP sequencing data obtained using GBTS technology, consecutive SNPs were examined in a sliding window of 15 SNPs. The ratio between the numbers of SNPs from the Y476 and Nip genomes was calculated in each window. Based on the allele ratio (Nip/Y476), each window genotype was then defined as having a homozygous Nip genotype (larger than 11:4) or a homozygous Y476 genotype (smaller than 2:13). In this way, we identified the recombination breakpoints in all lines, after which we aligned all chromosomes of all lines and compared them in intervals of at least 100 kb. Adjacent 100-kb intervals with the same genotype across the entire population were recognized as a single recombination bin. Finally, 1542 recombination bins were generated for CSSL/Nip population genotyping. By using a same method, 1075 recombination bins were generated for CSSL/9311 population genotyping.

## Phenotype identification and QTLs mapping

The phenotypes of nine agronomic traits were investigated in three environments (Supplementary Tables 17 and 23). To assess salt tolerance, rice seeds were rinsed with sterile water, placed on filter paper soaked in water, and allowed to germinate for 2 days at 28 °C. The germinated seedlings were grown in a culturing room at 28 °C, with a photoperiod of 12 h and 70% humidity. The nutrient solutions were changed every 4 days, and the seedlings were treated with NaCl solutions at the selected time points. For Nip and Y476 salt tolerance experiments, 35-day-old seedlings of Nip were transferred to nutrient soil, and wild rice Y476 seedlings with the same growth vigor were transferred to the same soil. After growing in the soil for 10 days, Nip and Y476 were treated with 150 mM NaCl for 15 days and photographed. For CSSL/Nip population salt tolerance experiments, all CSSLs were planted in a salt stress pool, with 85 mM (0.5%) salt stress treatment during the whole growth period. The salt tolerance level of each line was evaluated after heading. All experiments were repeated three times independently, and analyses were performed based on established evaluation criteria for rice germplasm[75]. QTLs for each trait were identified with QTL IciMapping 4.1[76]. The RSETP-LRT-ADD mapping method was applied with a logarithm of odds (LOD) threshold of 2.5.

## RNA isolation and qRT-PCR

Total RNA was extracted using the RNA Easy Fast Plant Tissue Kit (TIANGEN, Beijing, China) according to the manufacturer's protocols. Reverse transcription (RT) reactions were performed with the Prime-Script™ RT reagent Kit with gDNA Eraser (TaKaRa, Dalian, China) according to the manufacturer's instructions. Real-time qPCR experiments were performed using SuperReal PreMix Plus (SYBR Green) (TIANGEN, Beijing, China) on a CFX96 Real-Time PCR System (Bio-Rad, Beijing, China). The actin gene (*LOC_Os03g50885*) was used for the normalization of all qRT-PCR data. All qRT-PCR analyses were performed with at least three independent biological replicates. The $2^{-\triangle\triangle CT}$ method was used to calculate the relative expression levels

with three technical replicates[77]. Primers are listed in Supplementary Data 7.

## RNA-seq analysis

Total RNA was isolated from the leaves of wild rice Y476, Nip and N154 at 0 h, 24 h, and 48 h after inoculating rice blast fungus (three biological replicates). Leaves of Nip and N133 were collected at 0 h, 12 h, and 24 h after treatment with 200 mM (1.16%) NaCl. RNA-seq data were then analyzed[78]. The reads were mapped to the rice reference genome (MSUv7)[79] using Hisat2[62]. Calculation of read counts and DEG identification were performed using DEseq2[78,80]. Genes with an expression | log2Fold Change | >= 1 and FDR < 0.05 were defined as significantly differentially expressed. GO enrichment analysis was performed using PlantGSEA[81] and AgriGO[82].

## Fungus inoculation

*Magnaporthe oryzae* isolates were grown on oatmeal agar for about two weeks at 28 °C before producing spores. Conidia were induced under light for 2–3 days, and spores were collected in sterile water with Tween 20. In the experimental field, the blast resistance of rice plants was determined by transplanting the diseased plants as spreader lines. In the laboratory, rice blast resistance was tested by spray inoculation and punch inoculation. For punch inoculation, 4 μL of the spore suspension was pipetted at three spots on each leaf, and leaves were kept in a culture dish containing 0.1% 6-benzoaminopurine sterile water to keep them moist. Lesion length was measured 5–7 days post-inoculation[14]. All inoculation experiments were repeated three times independently.

## Statistical analysis

We used SPSS for statistical analysis. Data are mean ± SD. Two-sided Student's $t$ test was used to analyze the significant difference between two groups. A two-sided hypergeometric test was employed to analysis the GO enrichment, which were made for multiple comparisons by the Benjamini–Hochberg procedure to control the False Discovery Rate (FDR).

## Reporting summary

Further information on research design is available in the Nature Portfolio Reporting Summary linked to this article.

## Data availability

The raw sequencing data and genome assembly have been deposited in the National Center for Biotechnology Information (NCBI) under the Bioproject PRJNA1029807 and the National Genomics Data Center (NGDC) under the Bioproject PRJCA015108. The genome assembly and annotation are also available in figshare [https://figshare.com/articles/dataset/Genome_sequence_and_annotation_of_Hap1_Hap2_and_primary_of_Y476/24798696]. Source data are provided with this paper.

## Code availability

The code used for this paper is available at Zenodo [https://zenodo.org/records/10792568].

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

## Acknowledgements

This work was supported by the National Key R&D Program of China (2021YFD1200100 and 2021YFD1200501), Hainan Yazhou Bay Seed Laboratory (a project of B21HJ0215), and the Agricultural Science and Technology Innovation Program of the Chinese Academy of Agricultural Sciences.

## Author contributions

W.Q. and H.H. conceived and designed the experiments. J.H. and Y.Z. performed most of the experiments. Y.L. performed the rice blast resistance identification and transgenic experiments. M.X., S.W., Y.N., and Yanyan W. conducted phenotype investigation. C.L. provided rice blast strains. M.Z., X.X., W.F., Z.L., W.G., L.Z., and Y.C. carried out rice sowing and transplanting work. Z.H. and X.Z. provided guidance for this study. X.S. conducted a salt tolerance assessment during the growth period. H.Z. and Yan W. led the bioinformatics analyses. Q.Y. and Q.Q. supervised the project. J.H., Y.Z., W.Q., and H.H. analyzed data and wrote the manuscript. All authors read and approved the manuscript.

## Competing interests

The authors declare no competing interests.
