## [Peer Review File · Nature Communications]

Haplotype-resolved gapless genome assembly and chromosome segment substitution lines facilitated gene identification in wild riceReviewers' Comments:

Reviewer #1:

Remarks to the Author:

The authors constructed a reference genome assembly of a wild rice, *O. rufipogon*, for characterizing structural differences (PAVs) against other two cultivated rice genome assemblies. Using an impressive 10-year production of chromosome segment substitution lines (CSSLs) through backcrossing with 9311 and Nip, the authors collected various quantitative traits, identifying one line (Y476) that showed the most favorable trait features, salt tolerance and rice blast resistance. The authors further identified potential candidate genes associated with such traits, and induced three Crisper/Cas-9 based knock-out lines to validate a blast-resistant gene LOC_Os07g35680. The knock-out lines showed matching expression level of itself and OsMADS26 to its counterpart in Nip, confirming that LOC_Os07g35680 is important for blast resistance in rice.

The authors claim their *O. rufipogon* genome assembly is a T2T assembly. I argue that their assembly is a pseudo-haplotype assembly, perhaps it can be called "high quality reference" at best, with a few more additional validation.

- The authors acknowledged the genome they assembled had a high level of heterozygosity, over 1%. It will be easier to assemble both haplotypes given their deep sequencing depth of HiFi and ONT Ultra-long (UL, >100k) reads. Collapsing and removing the other haplotype poses more issues at this level of heterozygosity, especially if trying to merge two different assemblies. I can see why the authors tried to polish 6 times; but it is not a recommended practice to apply such an excessive rounds of polishing on incomplete genome assemblies. Reads from the haplotype that is absent in the assembly will map to the pairing other haplotype allele in the assembly, eventually over-polish and create a mixed (mosaic) haplotype that does not exist in nature. This kind of structural / haplotype inconsistency is not going to get captured in k-mer based QV estimates. The small contigs the authors purged out are probably coming from the missing haplotype, which would contain potential genomic variations making the wild rice favorable to adapt.
- No read-level structural validation results are shown. The authors show 3 regions where the gaps were filled in Supplementary Fig. 4, however all show weak evidence that the reads agree to the polished consensus. There are many mismatches in panel A. No HiFi nor ONT read goes across the region in panel B. Also, it is unclear what C panel is representing. Why are there so many HiFi reads in white? By default, IGV shades reads by MQ, and white means there is a place elsewhere in the assembly the reads can align. How does the authors ensure the reads are mapped to the correct place? All panels in Supp Fig4 have too much noise in the ONT reads. I would like to see the same screenshots (especially panel C), after hiding any INDELS smaller than 10 bp in ONT reads. The authors should more actively use their "remarkable" ONT UL reads to prove structural consistency to claim T2T. Bionano is not able to cover centromeres, and is also sparse compared to ONT.
- Related to this, I am not convinced the PCR amplification and sequencing validates the inversions shown in Supp. Fig 7. Inversion breakpoints usually have a very similar inverted repeat structure. The authors need to show evidence that UL read maps span across the inversion breakpoint (inverted repeat) region. 60 bp Sanger sequencing reads are too short. Similar story goes to Supplementary Fig. 8.
- Read mapping % is not a recommended way to assess genome assembly accuracy. It is possible to map >95% of the reads on a collapsed assembly. How much of the assembly contains regions with even coverage?
- Five out of twelve chromosomes are missing telomeres at least on one end. How does the authors ensure the assembly reached to the end of the chromosome? Again, missing telomeres is counterintuitive for being "T2T".

It is not always guaranteed to have all gene copies captured "correctly" in an assembly, especially those that are known to have copy number variations. Both Y476 and Nip assemblies aren't complete in terms of haplotypes. What if the missing haplotype contained more copies (haplotype specific

duplications)? All claims regarding copy numbers should be validated by estimation from read coverage at minimum.

- Fig. 3D: Is it correct to claim Nip has only 1 copy? It is easier to assume higher copy numbers have a higher dosage effect, thus I would expect Nip to have much higher copy numbers. What if Nip was just collapsing this region, or not assembled, but more gene copies are present in the genome?

- Fig. 3F: Also here, confirm the copy numbers are the copies present in the genome. In addition, the expression level does not look significantly different.

A lot of the results would benefit from having a structurally reliable assembly. Given the datatypes they have, I could imagine generating a (near) haplotype resolved T2T diploid assembly. A haplotype resolved assembly will help finding more variants that better explain functional variations responsible to their eQTL results. At minimum, I would enquire the authors to confirm sequence and structural level accuracy for LOC_Os02g08540 and LOC_Os07g35680.

- What will be the mechanism of the two deletions around LOC_Os02g08540 affecting gene expression? Were there no base level differences between the two genes in Nip and Y476/N133? Likewise, how would the insertion in Y476/N154 LOC_Os07g35680 affect gene expression?

Why is it important to have salt tolerance? Perhaps a little more introduction could help understanding the motivation. Also, please use consistent units for salted water. Some places provide salt as %, some use mL. Provide the amount of total (salted) water along with the % NaCl, in a consistent way.

How was Or-I, Or-II, and Or-III classified? The PCA plot in Supp. Fig 1 shows the clusters were chosen arbitrary, not really showing evidence of the classification. The Admixture plot is also disagreeing with their chosen cluster (k). Y476 looks like a mixed breed between Aus and Indica.

How were the genome similarity measured? Are the Nip and 9311 at a comparable assembly quality? I wonder if this analysis is confounded by the assembly quality. There will be more or less PAVs found, depending on the underlying assembly. I would argue to remove any claims regarding PAVs unless the analysis was performed on some controlled region, which again, raises the question about missing haplotypes.

How were the significance measured? Provide statistical test and p-values for every expression level comparison in the main text. The authors claim in multiple places that the expression level "significantly changed" with no details.

Among traits, taste (stickiness, sugar, ...) is also an important measure. Did the two CSSL lines had similar taste traits close to Nip? Did the structural variants affect other characteristics linked to taste?

A few minor comments:

- The manuscript needs substantial improvements in English.

- The authors claim "The quality of long reads sequencing determines the quality of genome assembly, and the sequencing data generated in this study is particularly remarkable." I would suggest removing this sentence and let the readers judge for being "remarkable".

- Move the Illumina PE read (~62.2X) part up if this is the same data used for population structure analysis.

Reviewer #2:

Remarks to the Author:

The authors present a new assembly of wild rice and study several traits in cross lines of the wild and domesticated rice. Overall the work is of wide interest. I have focused my review on the assembly aspect since that is my area of expertise.

- There is no data availability statement so it's not clear where/how to get the raw data which makes it impossible to independently evaluate the quality of the assembly. While the final assembly was shared for review purposes, is it going to be made available in a public archive as well?

- The authors claim that their assembly is T2T, which implies it is complete and correct. While I have no doubt the assembly is more continuous than previous wild rice assemblies, there is very little evidence to support completeness and correctness. Based on my comments below, I do not agree that this assembly meets the telomere-to-telomere standard and should not be presented as such.

- The only completeness stats are based on BUSCO which measures core genes and says nothing about repetitive areas such as centromeres. The assembly is missing telomeres on 3 chromosome ends (5,9,10) and has no telomeres on chr11. It has no mitochondria or chloroplast either. There must be some missing sequence in those cases where the chromosome doesn't reach the telomere?

- Validation is based only on bionano alignments and short-read alignments. Neither of these can map accurately within large repetitive regions (like centromeres). There are long-read based validation tools such as VerityMap (<https://github.com/ablab/VerityMap>), Gavisunk (<https://github.com/pdishuck/GAVISUNK>), and the T2T-Polish pipelines (<https://github.com/arangrhie/T2T-Polish>) which check agreement of the assembly with the long reads to identify and correct or at least flag errors.

- The authors state that the genome has a high heterozygosity and start the assembly by producing a haplotype-resolved assembly via hifiasm + HiC. However, the ONT assembly likely collapsed both haplotypes so the gap fills and subsequent polishing may be introducing haplotype switch errors. These are not evaluated by the authors and should be (e.g. map all reads to assembly, call variants, and use HiC to link variants which can be compared to the assembly consensus). The missing sequence (from the other haplotype) could be functionally important but is not at all addressed in the current manuscript.

- There is insufficient details in the assembly section. The authors only state some contigs were removed and the genome was manually corrected. This should be detailed as to why the contigs could be removed (e.g. redundant with another region of the genome) and what corrections were made.

- Some of the methods are difficult to follow. For example, lines 158-159 say the "12 gap free chromosomes + 3 gaps". What does that mean? Are the gap-fill sequences considered separately? The mapping rate here is quoted as 99.9% but then in lines 188 the mapping rate is 99.15%. Both seem high given a single haplotype and >1% divergence between haplotypes. No details on mapping parameters are given so it's impossible to evaluate these claims.

- The exact N50 of the ONT reads should be given in the text (line 142). Supplementary Figure 3 also does not show the N50, it could have a dashed line corresponding to N50. It may also be more informative to show total bases in reads of a given length rather than number of reads to better show where the majority of sequencing bases are.

- Supplementary Figure 4 is not clear. I'm not sure what the reference is and the text is too small to read. There is no HiFi dropout in coverage so it's not clear why the HiFi assembly would have a gap. The indels in the ONT reads should be turned off (at least below 10-20bp) to make the plot more readable. I'm also not sure how to conclude that the gaps can be filled. The B panel has low ONT coverage and a sharp drop in the middle with no spanning HiFi reads. Whatever the reference is here seems to be inconsistent with the sequencing data.

- Supplementary Figure 5 doesn't say what light blue vs dark blue is. I presume these are regions where there is no bionano map alignments and thus would have no validation?

- Overall, the manuscript needs English proofreading and correction.
- No citation to RagTag
- No citation to MUMandCo

Reviewer #3:

Remarks to the Author:

The manuscript reported a near complete genome of wild rice Y476. And authors also identified new salt-resistant and disease-resistant genes by using genome PAVs and rice CSSLs. It will give rise to a general interest for researchers and public readers. However, some experiment designs and conclusions should be improved.

1. In line 291-297, a PAV was identified in a reported grain weight-related gene located in one QTL region. Then they claimed that "we believe that the existence of these PAVs may be the cause for significant differences in the expression of these genes and thus affect the phenotype". The authors presented one example to conclude a general description. If the authors found a SNP in a known gene, they may believe "SNPs may be the cause for significant difference". It should be more reasonable that they compare how many PAVs/SNP associated with gene expression variation, how many PAVs/SNP associated with gene function change by using a statistic method.
2. In Figure 1B, the manuscript did not describe the name of *M. oryzae* isolates. To my best knowledge, Nip is also resistant to some isolates of *M. oryzae*. In line 114, the manuscript should not claim that Y476 was immune to the rice blast pathogen. Y476 just was immune to the specific isolates of *M. oryzae* if the authors could not provide more evidence.
3. In Figure 1C, the growth environment and plant density are different in salt resistance evaluation. Please evaluate it in a more consistent condition.
4. In Figure 6 and Fig S15. The manuscript provided some evidence showing that LOC_Os02g08540 was a candidate gene in a QTL region of salt-resistance. Basic genetic proof is missing, and the manuscript should present the phenotype of LOC_Os02g08540 KO and OE in Nip or N133 background.
5. In Figure S17. The authors evaluated the rice blast resistance phenotype of CSSLs population by using mixed races/isolates of *M. oryzae*. And at least five QTL regions of rice blast resistance were identified. In Figure 7H, the manuscript showed that LOC_Os07g35680 KO resulted in resistance loss of N154 line. This result could arise two possibilities. First, LOC_Os07g35680 is R gene for the isolate of *M. oryzae* used in the experiment. Second LOC_Os07g35680 could affect the resistance of R gene in N154 background/Y476. The authors should get transformants of OrLOC_Os07g35860 in Nip and evaluate its phenotype.

Reviewer #4:

Remarks to the Author:

The genomes of wild rice provide large amounts of genes resource to improve agronomic, biotic resistance and abiotic tolerance traits in rice. In addition, the wild rice genomes are also used for studying plant genome evolution within a short timeframe. However, due to its relatively high heterozygosity, assembly of a high-quality reference genome of wild rice is quite difficult, which prevent harnessing the genetic diversity in *O. rufipogon* for rice improvement. The manuscript by Huang et al reported the first telomere-to-telomere (T2T) genome assembly and annotation for one typical Chinese *O. rufipogon* accession Y476 with improvements in contiguity, completeness and correctness. In addition, they constructed two CSSL population and identified a large amount of QTLs associated with agronomic, biotic and abiotic stresses. The authors provided lot of experiment evidences to support their findings. However, they need to address and response the following questions before its possible publication.

1. In Fig 1C, how many days after the seed germination when the salt treatment began? This need to be added in the legend. In addition, the scale bars need to be added in Fig 1B and 1C.
2. Related references need to be cited in line 229-231 and line 242-243.
3. Whether LOC_Os02g08540 and LOC_Os07g35680 under selective sweep during the domestication of *O. rufipogon* to cultivated rice?
4. Does the alleles of LOC_Os02g08540 and LOC_Os07g35680 from Y476 exist in other rice cultivars and accessions? In addition, phylogenetic analysis of LOC_Os02g08540 is suggested to perform.
5. In Supplemental Fig 14, the grains of N214 and Nip need to be shown. In addition, the panels need to be labeled by ABC.
6. In Supplemental Fig 16, given that the 1,000-grain weight, grain length and grain number per plant of N133 are higher than that of Nip, how about the yield per plant of Nip and N133?
7. A title need to be added in Supplemental Fig 17. The scale bar need to be added in Supplemental Fig 17B.
8. The characters of gene names in Fig 6G, 7C-7E, Supplemental Fig 14, Supplemental Fig 15D-15G, Supplemental Fig 17C-17D and Supplemental Fig 20 should be italic.

RESPONSE TO REVIEWER COMMENTS

Reviewer #1 (Remarks to the Author):

The authors constructed a reference genome assembly of a wild rice, *O. rufipogon*, for characterizing structural differences (PAVs) against other two cultivated rice genome assemblies. Using an impressive 10-year production of chromosome segment substitution lines (CSSLs) through backcrossing with 9311 and Nip, the authors collected various quantitative traits, identifying one line (Y476) that showed the most favorable trait features, salt tolerance and rice blast resistance. The authors further identified potential candidate genes associated with such traits, and induced three Crisper/Cas-9 based knock-out lines to validate a blast-resistant gene *LOC_Os07g35680*. The knock-out lines showed matching expression level of itself and *OsMADS26* to its counterpart in Nip, confirming that *LOC_Os07g35680* is important for blast resistance in rice.

The authors claim their *O. rufipogon* genome assembly is a T2T assembly. I argue that their assembly is a pseudo-haplotype assembly, perhaps it can be called “high quality reference” at best, with a few more additional validations.

1. The authors acknowledged the genome they assembled had a high level of heterozygosity, over 1%. It will be easier to assemble both haplotypes given their deep sequencing depth of HiFi and ONT Ultra-long (UL, >100k) reads. Collapsing and removing the other haplotype poses more issues at this level of heterozygosity, especially if trying to merge two different assemblies. I can see why the authors tried to polish 6 times; but it is not a recommended practice to apply such an excessive rounds of polishing on incomplete genome assemblies. Reads from the haplotype that is absent in the assembly will map to the pairing other haplotype allele in the assembly, eventually over-polish and create a mixed (mosaic) haplotype that does not exist in nature. This kind of structural / haplotype inconsistency is not going to get captured in k-mer based QV estimates. The small contigs the authors purged out are probably

coming from the missing haplotype, which would contain potential genomic variations making the wild rice favorable to adapt.

Your valuable comments are appreciated. To answer your questions and improve the reliability of our results, we have reassembled the Y476 haplotype genome using hifiasm (v0.18.4) with the parameters -h1 -h2 -ul. A novel *de novo* assembly strategy was implemented, integrating HiFi, ONT, and Hi-C data, yielding a refreshed primary genome and hap1/hap2 genomes. The novel assembly approach, which directly integrates two types of sequencing data, resulted in two haplotype-resolved genomes of greater continuity (Supplementary Fig. 5). While the previous version of the primary genome contained three gaps, the newly updated primary genome contains twelve gap-free chromosomes. Adhering to your suggestion, we refrained from over-polishing. Subsequently, we employed HiC data to anchor the contigs into twelve chromosomes and appraise their quality. The HiC interaction heatmap is shown in Figure 2B. The quality evaluation results and annotation of the genomes are provided in Table 1. Furthermore, any claim to a “T2T genome” within the article has been revised to a “haplotype gapless genome”.

2. No read-level structural validation results are shown. The authors show 3 regions where the gaps were filled in Supplementary Fig. 4, however all show weak evidence that the reads agree to the polished consensus. There are many mismatches in panel A. No HiFi nor ONT read goes across the region in panel B. Also, it is unclear what C panel is representing. Why are there so many HiFi reads in white? By default, IGV shades reads by MQ, and white means there is a place elsewhere in the assembly the reads can align. How does the authors ensure the reads are mapped to the correct place? All panels in Supp Fig4 have too much noise in the ONT reads. I would like to see the same screenshots (especially panel C), after hiding any INDELS smaller than 10 bp in ONT reads. The authors should more actively use their “remarkable” ONT UL reads to prov structural consistency to claim T2T. Bionano is not able to cover centromeres, and is also sparse compared to ONT.

We sincerely apologize for the ambiguity in the previous Supplementary Fig. 4. Following your earlier suggestions, after reassembling the genome, we re-evaluated the three regions, removed S-

Fig.4 and generated the figure included below (Response Fig. 1). Purple and black denote indels; we concealed short indels and expanded the length of the displayed area. The results demonstrate good alignment and coverage rates of HiFi and ONT data within the three areas (500-700Kb).

In addition, we randomly validated some gap regions previously present on the wild rice genome. These gaps ranged from 3782 to 5178 bp and were assessed by PCR amplification and sequencing. All of the gaps were confirmed (Supplementary Fig. 8C).

Response Fig. 1. IGV depiction of HiFi and UL read mapping to the Y476 genome across three gap regions.

(A-C) The coverage depth and read mapping status of HiFi data at upstream and downstream gap positions are shown above, and the coverage depth and read mapping status of UL reads are shown below. Gap regions in Chr.5 (A), Chr.8 (B), and Chr.12 (C) are shown.

3. Related to this, I am not convinced the PCR amplification and sequencing validates the inversions shown in Supp. Fig 7. Inversion breakpoints usually have a very similar inverted repeat structure. The authors need to show evidence that UL read maps span across the inversion breakpoint (inverted repeat) region. 60 bp Sanger sequencing reads are too short. Similar story goes to Supplementary Fig. 8.

We appreciate your suggestions. For the validation of inversions, we aligned ultra-long (UL) read and Hi-C data back to the inversion breakpoints, as depicted in Supplementary Fig. 7. Notably, the UL reads align well in the Y476 genome breakpoint area, and the Hi-C heatmap of Chromosome 6 between Nipponbare and Y476 displays an inversion. Moreover, regarding the validation of the inversion, our Sanger sequencing read length was greater than 600 bp (Supplementary Fig. 7E).

To validate the structural variations shown in Supplementary Fig. 8, we aligned UL reads to the Nipponbare genome, revealing sizable structural variations in these regions, as depicted in Response Fig. 2.

Response Fig. 2. UL read maps span across the SV regions.

(A-E) Overview of five SV and UL read maps IGV. The panels show the five SVs validated by PCR (Supplementary Fig. 8) located on Chr.2 (A), Chr.6 (B), Chr.7 (C), Chr.9 (D), and Chr.12 (E).

4. Read mapping % is not a recommended way to assess genome assembly accuracy. It is possible to map >95% of the reads on a collapsed assembly. How much of the assembly contains regions with even coverage?

We appreciate your guidance. To evaluate the uniformity of coverage across the genome, we deployed HiFi, ONT, and WGS data. As delineated in Response Table 1, for HiFi reads, approximately 98.84% of regions demonstrate coverage depth within the 28X-108X range, while about 1.16% of the area exhibits non-uniform coverage depth. For ONT reads, approximately 99.51% of regions demonstrate coverage depth within the 11X-61X range, while about 0.49% of the area exhibits non-uniform coverage depth. For WGS reads, approximately 98.75% of regions demonstrate coverage depth within the 20X-80X range, while about 1.25% of the area exhibits non-uniform coverage depth.

5. Five out of twelve chromosomes are missing telomeres at least on one end. How does the authors ensure the assembly reached to the end of the chromosome? Again, missing telomeres is counterintuitive for being “T2T”

Thank you for your insightful recommendations. In compliance with your previous advice, the telomere status of our reassembled genomes is outlined in Response Table 2. Although we've progressed from missing five telomeres to a mere two, we still lack the sequences preceding the telomeres of chromosome 9 and 10. Consequently, any claim to a T2T genome within the article has been revised to a “haplotype gapless genome”.

6. It is not always guaranteed to have all gene copies captured “correctly” in an assembly, especially those that are known to have copy number variations. Both Y476 and Nip assemblies aren't complete in terms of haplotypes. What if the missing haplotype contained more copies (haplotype specific duplications)? All claims

regarding copy numbers should be validated by estimation from read coverage at minimum.

We greatly appreciate your queries. Broadly speaking, to determine gene copy numbers for the Nip assembly, we downloaded the T2T-Nip genome (<http://www.ricesuperpir.com/web/download>). To validate gene copy numbers in Y476, in accordance with your suggestion, we used WGS, HiFi, and ONT to assess the read coverage depth for each gene. The results are exhibited in Response Table 3.

7. Fig. 3D: Is it correct to claim Nip has only 1 copy? It is easier to assume higher copy numbers have a higher dosage effect, thus I would expect Nip to have much higher copy numbers. What if Nip was just collapsing this region, or not assembled, but more gene copies are present in the genome?

We utilized the T2T-Nip genome to examine whether the *LOC_Os04g32350* gene possessed additional copies, with the result indicating two copies of this gene, implying that Y476 still maintains a greater number of copies. Of note, this duplication event occurred 7M upstream from the *LOC_Os04g32350* gene, which could potentially account for the inverse dosage effect. Concurrently, we used read coverage depth to validate the copy number in Y476.

8. Fig. 3F: Also here, confirm the copy numbers are the copies present in the genome. In addition, the expression level does not look significantly different.

We utilized the T2T-Nip genome to examine whether the NBS-LRR gene from Nip (Fig. 3F) had additional copies, with the result indicating three copies. For Y476, we validated the gene copy number using read coverage depth. Simultaneously, we conducted a significance test for the expression level; a two-tailed Student's t-test was used to analyze the expression of Y476 (panel D and F in Figure 3). The results showed that there was a significant difference between Y476 and Nip (p-value in panel D is 0.00014; p-value in panel F is <0.0001). Thanks for your valuable comments. We have noted the p-value in the new manuscript.

9. A lot of the results would benefit from having a structurally reliable assembly. Given the datatypes they have, I could imagine generating a (near) haplotype resolved T2T diploid assembly. A haplotype resolved assembly will help finding more variants that better explain functional variations responsible to their eQTL results. At minimum, I would enquire the authors to confirm sequence and structural level accuracy for LOC_Os02g08540 and LOC_Os07g35680.

We have confirmed the sequence variations of *LOC_Os02g08540* and *LOC_Os07g35680* by genomic DNA sequencing, and the SVs were also verified by PCR.

We amplified the CDS region and promoter region of the *LOC_Os07g35680* gene from Y476 by PCR and sequenced the amplified products by Sanger sequencing. The sequencing results were consistent with the assembly sequence of the Y476 haplotype genome. Based on the assembled Y476 haplotype genome, the two haplotype sequences of the *LOC_Os07g35680* gene and the Nip sequence were compared. No SV was present between Hap1 and Nip, whereas a large SV was revealed in the intron between Hap2 and Nip, with a length of 7.8 kb. To verify the presence of the SV between Hap2 and Nip, PCR amplification and sequencing were performed based on both ends of the SV sequence, and the results confirmed the SV.

Similarly, we amplified the CDS region and promoter region of the *LOC_Os02g08540* gene from Y476 by PCR and sequenced the amplified products by Sanger sequencing. The sequencing results were consistent with the assembly sequence of the Y476 haplotype genome. Based on the assembled Y476 haplotype genome, the two haplotype sequences of the *LOC_Os02g08540* gene and the Nip sequence were compared. An 88 bp SV was found in the promoter of the gene in Nip and Y476.

10. What will be the mechanism of the two deletions around LOC_Os02g08540 affecting gene expression? Were there no base level differences between the two genes in Nip and Y476/N133? Likewise, how would the insertion in Y476/N154 LOC_Os07g35680 affect gene expression?

Excellent questions!

In this manuscript, we focus on two unknown genes. *LOC_Os02g08540* confers salt tolerance and *LOC_Os07g35680* confers rice blast resistance. SVs were found in the promoter of *LOC_Os02g08540* and on an intron of *LOC_Os07g35680*, and their expression levels were significantly changed in the CSSL.

There are also many base level differences between the two *LOC_Os02g08540* alleles of Nip and Y476. The candidate cis-acting elements are listed in Response Table 4. The different cis-acting elements in the two wild rice alleles may have led to differences in expression.

Transgenic verification is the best method of understanding the mechanisms by which SVs affect gene expression. We have already performed such transformations using rice, but the generation of transgenic plants requires at least 1 year. We have selected near-isogenic lines (NILs) of these two genes in the CSSL/9311 population, because the CSSL/9311 population is an advanced backcrossed generation population. SVs and expression were confirmed for these two wild rice alleles, and the phenotypes of the NILs were identified. The results of these analyses are provided in the revised manuscript as Supplemental Figure 19 and Supplemental Figure 22. In addition, we would like to mention a previous study (Shang et al. 2022, Cell Research, 32, 878-896), in which SVs were found to affect agronomic traits by altering gene expression and NILs were used to confirm gene function. Therefore, we believe that the evaluation of NILs for our selected genes represents evidence that verifies the effects of SVs on gene expression and phenotypes in our study.

11. Why is it important to have salt tolerance? Perhaps a little more introduction could help understanding the motivation. Also, please use consistent units for salted water. Some places provide salt as %, some use mL. Provide the amount of total (salted) water along with the % NaCl, in a consistent way.

Cultivated rice is a salt-sensitive plant, and salt stress is one of the main limiting factors in rice production, so the discovery of salt tolerance alleles in wild rice genomes is an important task that can support rice breeding efforts. We have provided some background information in the Introduction section. Thank you very much for your suggestions.

For rice salt tolerance experiments, we generally use “mM NaCl” as the unit for laboratory experiments. For salt treatment in rice paddy fields, we usually use “%” as the unit. However, we agree that consistent units would be clearer, so we have used “mM” for all salt concentrations in the revised manuscript.

12. How was Or-I, Or-II, and Or-III classified? The PCA plot in Supp. Fig 1 shows the clusters were chosen arbitrary, not really showing evidence of the classification. The Admixture plot is also disagreeing with their chosen cluster (k). Y476 looks like a mixed breed between Aus and Indica.

Huang (Huang et al. 2012, Nature 490, 497-501) investigated the population structure of 446 *O. rufipogon* accessions. Based on the results of neighbor-joining tree construction and principal component analysis (PCA), they classified the *O. rufipogon* species into three types, designated as Or-I, Or-II and Or-III, and they found that the *O. rufipogon* population structure was strongly correlated with geographic distribution. Based on Huang's classification method, we also divided the *O. rufipogon* population into three groups: Or-I, Or-II and Or-III. In this study, we explored the genome of Y476 by three methods: phylogenetic tree analysis, PCA and admixture analysis. The phylogenetic tree results and admixture results support the classification of Y476 as Or-II. We acknowledge that the PCA results showed that Y476 did not have a clear attribution. To be cautious, we conducted PCA analysis based on only 446 wild rice samples (Response Fig. 3), and the results were consistent with the previous results showing that Y476 is clustered most closely to Or-II. Thus, based on the comprehensive results of the phylogenetic tree analysis, admixture analysis and PCA, we have classified Y476 into the Or-II group.

Response Fig. 3. Genetic structure analysis of Y476.

Principal component analysis plots of the wild rice Y476 indicated that it was most closely related to the Or-II *O. rufipogon* population. PC1 and PC2 represent the first and second eigenvectors, respectively.

13. How were the genome similarity measured? Are the Nip and 9311 at a comparable assembly quality? I wonder if this analysis is confounded by the assembly quality. There will be more or less PAVs found, depending on the underlying assembly. I would argue to remove any claims regarding PAVs unless the analysis was performed on some controlled region, which again, raises the question about missing haplotypes.

We wholeheartedly concur with your advice. By downloading the T2T Nip genome, we assessed the quality of the three genomes (Response Table 5). The genomic continuity of Y476 is comparable to that of T2T-Nip, and it is 2.7 times greater than that of 9311. In terms of completeness, as assessed by genome BUSCO, LAI, and the number of telomeres, Y476 and T2T-Nip significantly surpass

9311. Y476 has a BUSCO score of 98.80%, an LAI of 24.69, and 22 telomeres; T2T-Nip has a BUSCO score of 98.80%, an LAI of 22.11, and 24 telomeres; whereas 9311 has a BUSCO score of 98.70%, an LAI of 21.99, and a single telomere. Given that the Nip and Y476 genomes were assembled using HiFi or UL data and are deemed to be of equivalent quality in terms of identifying large structural variations, we revisited the identification and analysis of SV, and amendments were made to the original text accordingly. Simultaneously, we have expunged any assertions concerning the 9311 PAV and acknowledged that this may result in missing some SVs in our manuscript.

14. How were the significance measured? Provide statistical test and p-values for every expression level comparison in the main text. The authors claim in multiple places that the expression level “significantly changed” with no details.

We sincerely apologize for any confusion caused by the insufficient description and tagging in the details. Thank you for your thoughtful suggestion. The significance analysis in the manuscript was performed using a two-tailed Student’s t-test, and P-values were provided in the revision.

15. Among traits, taste (stickiness, sugar, ...) is also an important measure. Did the two CSSL lines had similar taste traits close to Nip? Did the structural variants affect other characteristics linked to taste?

Very good question!

Taste traits, including amylose content, stickiness, and protein content, among others, are very important traits for modern rice breeding. The genetic material of both CSSL populations assessed in this manuscript, as innovative rice germplasms useful for breeding, were distributed to our colleagues for studies of target taste traits. The CSSLs assessed in our study had large phenotypic variations in taste traits. Thank you for your inquiry regarding taste traits. Your valuable suggestions have provided new directions and ideas for our future studies. We will conduct some rice quality and taste studies based on two sets of CSSLs, and we hope to find novel genes from wild rice that can improve rice taste.

A few minor comments:

16. The manuscript needs substantial improvements in English.

We apologize for the language problems in the original manuscript. The manuscript was improved with assistance from a native English speaker.

17. The authors claim “The quality of long reads sequencing determines the quality of genome assembly, and the sequencing data generated in this study is particularly remarkable.” I would suggest removing this sentence and let the readers judge for being “remarkable”.

Sincerely thank you for your suggestion, We have removed this sentence in the new manuscript.

18. Move the Illumina PE read (~62.2X) part up if this is the same data used for population structure analysis.

Thank you for your valuable suggestion. We have made modifications in the revised version according to your suggestions

Reviewer #2 (Remarks to the Author):

The authors present a new assembly of wild rice and study several traits in cross lines of the wild and domesticated rice. Overall, the work is of wide interest. I have focused my review on the assembly aspect since that is my area of expertise.

1. There is no data availability statement so it's not clear where/how to get the raw data which makes it impossible to independently evaluate the quality of the assembly. While the final assembly was shared for review purposes, is it going to be made available in a public archive as well?

We greatly appreciate your constructive comments. The raw sequence data reported in our study have been deposited in the Genome Sequence Archive (Genomics, Proteomics & Bioinformatics 2021) of the National Genomics Data Center (Nucleic Acids Res 2022), China National Center for Bioinformation/Beijing Institute of Genomics, Chinese Academy of Sciences (BioProject: PRJCA015108) and are publicly accessible at

<https://ngdc.cnbc.ac.cn/bioproject/browse/PRJCA015108>. The genome assembly and annotation files are available at: https://figshare.com/articles/dataset/The_genome_assembly_and_annotation_of_Y476/23796624.

2. The authors claim that their assembly is T2T, which implies it is complete and correct. While I have no doubt the assembly is more continuous than previous wild rice assemblies, there is very little evidence to support completeness and correctness. Based on my comments below, I do not agree that this assembly meets the telomere-to-telomere standard and should not be presented as such.

We appreciate your advice, and we also acknowledge that some gaps exist in comparison to the T2T assembly. We have reassembled the Y476 haplotype genome using hifiasm (v0.18.4) with the parameters -h1 -h2 -ul. A novel de novo assembly strategy was implemented, integrating HiFi, ONT, and Hi-C data, which yielded a refreshed primary genome and Hap1/Hap2 genomes. Adhering to your suggestion, we refrained from over-polishing. Subsequently, we employed HiC data to anchor the contigs into twelve chromosomes and appraise their quality. The HiC interaction heatmap is shown in Figure 2. The quality evaluation and annotation results for the genomes are provided in Table 1. Furthermore, any claim to a T2T genome within the article has been revised to a “haplotype gapless genome”.

3. The only completeness stats are based on BUSCO which measures core genes and says nothing about repetitive areas such as centromeres. The assembly is missing telomeres on 3 chromosome ends (5,9,10) and has no telomeres on chr11. It has no mitochondria or chloroplast either. There must be some missing sequence in those cases where the chromosome doesn't reach the telomere?

We sincerely appreciate your suggestions. By using HiFi and UL data, reassembled the genome, which now lacks only two telomeres. Subsequently, we assessed the completeness of repetitive regions, including centromeres and telomeres. We employed the number and total length of rice telomere/centromere repeat units to evaluate the completeness of the telomere/centromere regions and compared the results for Y476, Nip-T2T (unpublished), gap-free MH63, and gap-free ZS97, which represent the highest quality rice genomes that are currently available. The results reveal that

the integrity of Y476's telomeres is second only to that of Nip-T2T. The number of CentO units in Y476's centromeres is relatively complete, and Nip-T2T is the most complete genome available in this regard (Response Table 6). Finally, we reassembled the chloroplast and mitochondrial genomes of Y476, which measured 134,551 bp and 584,028 bp in length, respectively. However, we still lack the sequence preceding the telomeres of chromosomes 9 and 10 for Y476.

4. Validation is based only on bionano alignments and short-read alignments. Neither of these can map accurately within large repetitive regions (like centromeres). There are long-read based validation tools such as VerityMap (<https://github.com/ablab/VerityMap>), Gavisunk (<https://github.com/pdishuck/GAVISUNK>), and the T2T-Polish pipelines (<https://github.com/arangrhie/T2T-Polish>) which check agreement of the assembly with the long reads to identify and correct or at least flag errors.

We utilized the VerityMap (Mikheenko et al., 2020, *Bioinformatics* 36, i75-i83) and T2T-Polish (McCartney et al., 2022, *Nature methods* 19, 687-695) pipelines to assess the quality of our novel genome, and the errors identified in the results are shown in Response Table 7 and Response Table 8. The VerityMap results reveal a combined length of possible heterozygous sites and errors amounting to 15.47 Mb, while the low-quality areas identified by T2T-Polish total 6.30 Mb.

References:

Mikheenko, A. et al. TandemTools: mapping long reads and assessing/improving assembly quality in extra-long tandem repeats. *Bioinformatics* 36, i75-i83 (2020).

McCartney, A.M. et al. Chasing perfection: validation and polishing strategies for telomere-to-telomere genome assemblies. *Nature methods* 19, 687-695 (2022).

5. The authors state that the genome has a high heterozygosity and start the assembly by producing a haplotype-resolved assembly via hifiasm + HiC. However, the ONT assembly likely collapsed both haplotypes so the gap fills and subsequent polishing may be introducing haplotype switch errors. These are not evaluated by the authors and

should be (e.g. map all reads to assembly, call variants, and use HiC to link variants which can be compared to the assembly consensus). The missing sequence (from the other haplotype) could be functionally important but is not at all addressed in the current manuscript.

We greatly appreciate your insightful comments. To address your questions and enhance the reliability of our results, we have assessed haplotype switch errors on the newly assembled Hap1 and Hap2 genomes mentioned above. We employed a method that has been used previously for evaluating the haploid genome of *Camellia sinensis* with HiC data (Zhang et al. 2021 Nat. Genet. 53, 1250-1259). The results indicate a switch error rate of 7.1% for the wild rice, which is similar to that of *C. sinensis* (5.9%).

6. There is insufficient details in the assembly section. The authors only state some contigs were removed and the genome was manually corrected. This should be detailed as to why the contigs could be removed (e.g. redundant with another region of the genome) and what corrections were made.

Regarding the 254 (16.2 Mb) unanchored contigs, 100% are short sequences (< 200 kb) and all were aligned to chromosomes with high similarity, yet they only cover approximately 0.6 M of the genome. These results suggest that these sequences are either repetitive or redundant.

The principal corrections involved adjusting the synteny with published rice genome chromosomes. The synteny of our HiC-anchored genome with the rice genome is depicted in Response Fig. 3A, and the ultimate synteny with the rice genome is shown in Response Fig. 3B. Importantly, this corrective step did not alter the genome sequence.

Response Fig. 3. Collinearity between the Y476 genome before and after correction with the Nip genome. (A) *De novo* assembled contig ID correspondence to reference Nip genome Chr IDs. The X-axis indicates the chromosome of the Nip genome, and the y-axis indicates the *de novo* assembled contigs. **(B)** Correction of Y476 genome contig ID correspondence to reference Nip genome Chr IDs. The X-axis indicates the chromosome of the Nip genome, and the y-axis indicates the *de novo* assembled contigs.

7. Some of the methods are difficult to follow. For example, lines 158-159 say the "12 gap free chromosomes + 3 gaps". What does that mean? Are the gap-fill sequences considered separately? The mapping rate here is quoted as 99.9% but then in lines 188 the mapping rate is 99.15%. Both seem high given a single haplotype and >1% divergence between haplotypes. No details on mapping parameters are given so it's impossible to evaluate these claims.

We profoundly apologize for any confusion caused by our unclear descriptions. In this instance, we utilized ONT ultra-long reads to fill three gaps that in the HiFi data assembly. Ultimately, the alignment rate for these three gap regions, using Illumina sequencing data, was 99.9%. On line 188, the alignment rate for the whole genome was 99.15%. Both rates were derived using the same parameters; alignments were performed in BWA-MEM using default parameters, and alignment rates and coverage were calculated in Qualimap2 using default parameters. We have elaborated on our novel assembly procedure and parameters above.

8. The exact N50 of the ONT reads should be given in the text (line 142). Supplementary Figure 3 also does not show the N50, it could have a dashed line corresponding to N50. It may also be more informative to show total bases in reads of

a given length rather than number of reads to better show where the majority of sequencing bases are.

We are immensely grateful for your advice. Amendments have been made to both the main text and accompanying figures, the results of which are displayed in the new Supplementary Fig. 3. The exact N50 values of the ONT reads are included in the revised manuscript.

9. Supplementary Figure 4 is not clear. I'm not sure what the reference is and the text is too small to read. There is no HiFi dropout in coverage so it's not clear why the HiFi assembly would have a gap. The indels in the ONT reads should be turned off (at least below 10-20bp) to make the plot more readable. I'm also not sure how to conclude that the gaps can be filled. The B panel has low ONT coverage and a sharp drop in the middle with no spanning HiFi reads. Whatever the reference is here seems to be inconsistent with the sequencing data.

We sincerely apologize for the ambiguity in the previous Supplementary Fig. 4. Following your earlier suggestions, after reassembling the genome, we re-evaluated the three regions and replaced the previous S-Fig.4 with **Response Fig. 4**. Purple and black denote indels; we concealed short indels and expanded the length of the displayed area. The results demonstrate good alignment and coverage rates for the HiFi and ONT data within the three areas (500-700Kb).

In addition, we randomly validated some gap regions previously present on the wild rice genome. These gaps ranged from 3782 to 5178 bp, and all gaps were verified by PCR amplification and sequencing (**Supplementary Fig. 8C**).

Response Fig. 4. IGV depiction of HiFi and UL read mapping to the Y476 genome across three gap regions.

(A-C) The coverage depth and read mapping status of HiFi data at upstream and downstream gap positions are shown above, and the coverage depth and read mapping status of UL reads are shown below. Gap regions in Chr.5 (A), Chr.8 (B), and Chr.12 (C) are shown.

10. Supplementary Figure 5 doesn't say what light blue vs dark blue is. I presume these are regions where there is no bionano map alignments and thus would have no validation?

Thank you for your suggestion. The yellow labels denote areas of non-alignment, whilst light blue indicates regions devoid of Bionano enzyme sites. The dark blue, conversely, indicates the presence of enzyme sites with successful alignment.

11. No citation to RagTag.

Thank you for underlining this deficiency. In previous manuscripts, we applied ragtag, but ignored the citation. In the new version of the manuscript, we have reassembled Y476 haplotype genome, and RagTag was not applied in the new assembly process, therefore its citation was not added in the new manuscript.

12. No citation to MUMandCo.

Thank you for the suggestion. Citation to MUMandCo was added in the corresponding positions in the revised manuscript.

Reviewer #3 (Remarks to the Author):

The manuscript reported a near complete genome of wild rice Y476. And authors also identified new salt-resistant and disease-resistant genes by using genome SVs and rice CSSLs. It will give rise to a general interest for researchers and public readers. However, some experiment designs and conclusions should be improved.

1. In line 291-297, a PAV was identified in a reported grain weight-related gene located in one QTL region. Then they claimed that “we believe that the existence of these PAVs may be the cause for significant differences in the expression of these genes and thus affect the phenotype” . The authors presented one example to conclude a general description. If the authors found a SNP in a known gene, they may believe “SNPs may be the cause for significant difference” . It should be more reasonable that they compare how many PAVs/SNP associated with gene expression variation, how many PAVs/SNP associated with gene function change by using a statistic method.

We are very appreciative of your comments and constructive suggestions, and profoundly apologize for any confusion caused by our unclear descriptions. We have made the following change to address this issue: “Therefore, this SV may be the cause of the significant difference in the expression of the *OsSWEET14* gene in Y476 in comparison with Nip, and thus contribute to the grain weight phenotype.”

In the Results section “Global comparison and identification of unique gene families and novel genes of Y476”, we included the number of genes affected by SVs. According to your suggestion, we included the number of genes harboring SVs in their promoter region or coding region. Analysis of RNA-seq data from Nip and Y476 revealed 8,141 differentially expressed genes, among which

4,284 had at least one SV. Within this subset, 3,534 genes (82.49%) had at least one SV in the promoter region, while 1,899 (44.33%) had at least one SV in the coding sequence (CDS).

2. In Figure 1B, the manuscript did not describe the name of *M. oryzae* isolates. To my best knowledge, Nip is also resistant to some isolates of *M. oryzae*. In line 114, the manuscript should not claim that Y476 was immune to the rice blast pathogen. Y476 just was immune to the specific isolates of *M. oryzae* if the authors could not provide more evidence.

We greatly appreciate your insightful comments. We used a mix of *M. oryzae* isolates FJ07-5-2 and FJ07-8-1 to investigate blast resistance of wild rice and Nip in this study. We added information about the *M. oryzae* isolates to the legend of Figure 1 and Methods section, as well as in the main manuscript where these isolates were mentioned.

3. In Figure 1C, the growth environment and plant density are different in salt resistance evaluation. Please evaluate it in a more consistent condition.

We sincerely thank you for your valuable feedback. According to your suggestions, we have subjected Y476 and Nip to salt resistance treatment under more consistent conditions. Therefore, in the new manuscript, we have revised Figure 1C. Simultaneously, we have added a detailed description of the salt resistant treatment process in the Methods section.

For salt tolerance experiments, 45-day-old Nip seedlings were used for NaCl treatments. Because of wild rice's heterogeneity, seedlings from germinated wild rice seeds were not used for phenotype evaluation. We used wild rice germplasm with similar plant density and growth vigor for comparisons with the Nip seedlings and evaluated them in a consistent condition.

We repeated the salt tolerance experiments; representative wild rice individuals are shown in Figure 1C. Additionally, we also measured the salt tolerance of Y476 in a paddy field, which revealed that it can survive under 0.7% (120 mM NaCl) salt stress during its entire growth period. As shown in Supplementary Fig. 16C, Nip does not survive under 0.5% salt stress.

4. In Figure 6 and Fig S15. The manuscript provided some evidence showing that LOC_Os02g08540 was a candidate gene in a QTL region of salt-resistance. Basic genetic proof is missing, and the manuscript should present the phenotype of LOC_Os02g08540 KO and OE in Nip or N133 background.

Very good question.

We agree that much more genetic proof is required to verify *LOC_Os02g08540*. Efforts to generate plants with overexpression of *LOC_Os02g08540* are underway; however, the generation of homozygous transgenic plants requires more than one year. Therefore, in the present study, we used a near-isogenic line (NIL) generated on the 9311 background to confirm this gene's effect.

As we mentioned in the manuscript, another CSSL population generated on the *indica* "9311" background is an advanced generation backcross population. The corresponding NIL had salt tolerance better than that of 9311, as well as a higher expression level of *LOC_Os02g08540*. We believe that the evidence obtained using NILs provides evidence for this gene's effect. These results were added to the main manuscript as **Supplementary Figure 19**.

5. In Figure S17. The authors evaluated the rice blast resistance phenotype of CSSLs population by using mixed races/isolates of *M. oryzae*. And at least five QTL regions of rice blast resistance were identified. In Figure 7H, the manuscript showed that LOC_Os07g35680 KO resulted in resistance loss of N154 line. This result could arise two possibilities. First, LOC_Os07g35680 is R gene for the isolate of *M. oryzae* used in the experiment. Second LOC_Os07g35680 could affect the resistance of R gene in N154 background/Y476. The authors should get transformants of LOC_Os07g35860 in Nip and evaluate its phenotype.

We appreciate your comments.

As we mentioned above, a line with overexpression of this gene in the Nip background is in production, but it will require at least one year to generate. We used a NIL for this locus on the 9311 genetic background to confirm its function. The CSSL/9311 line C22 was selected as a NIL of this

gene, and its genotype, agronomic traits, yield related traits, and blast resistance were investigated using 9311 as the control. The expression pattern of *LOC_Os07g35680* in C22/9311 was similar to that of N154/Nip, and the blast resistance of C22 was better than that of 9311. Therefore, we believe that this evidence suggests that *LOC_Os07g35680* is a rice blast resistance gene. These results are included in the manuscript as Supplementary Figure 22.

In addition, we would like to mention a previous study (Shang et al. 2022, *Cell Research*, 32, 878-896), in which SVs were found to affect agronomic traits by altering gene expression and NILs were used to confirm gene function. Therefore, we believe that the evaluation of NILs for our selected genes represents evidence that verifies the effects of SVs on gene expression and phenotypes in our study. The two possibilities that you kindly point out in your comment are very important for our future studies of the function of this rice blast resistance gene and regulation networks. We are very appreciative!

Reviewer #4 (Remarks to the Author):

The genomes of wild rice provide large amounts of genes resource to improve agronomic, biotic resistance and abiotic tolerance traits in rice. In addition, the wild rice genomes are also used for studying plant genome evolution within a short timeframe. However, due to its relatively high heterozygosity, assembly of a high-quality reference genome of wild rice is quite difficult, which prevent harnessing the genetic diversity in *O. rufipogon* for rice improvement. The manuscript by Huang et al reported the first telomere-to-telomere (T2T) genome assembly and annotation for one typical Chinese *O. rufipogon* accession Y476 with improvements in contiguity, completeness and correctness. In addition, they constructed two CSSL population and identified a large amount of QTLs associated with agronomic, biotic and abiotic stresses. The authors provided lot of experiment evidences to support their findings. However, they need to address and response the following questions before its possible publication.

1. In Fig 1C, how many days after the seed germination when the salt treatment began? This need to be added in the legend. In addition, the scale bars need to be added in Fig 1B and 1C.

Thank you very much for your good suggestion.

For salt tolerance experiments, 45-day-old Nip seedlings were used for NaCl treatments. Because of wild rice's heterogeneity, seedlings from geminated wild rice seeds were not used for phenotype evaluation. We used wild rice germplasm with similar plant density and growth vigor for comparisons with the Nip seedlings and evaluated them in a consistent condition.

We repeated the salt tolerance experiments; representative wild rice individuals are shown in **Figure 1C**. Additionally, we also measured the salt tolerance of Y476 in a paddy field, which revealed that it can survive under 0.7% (120 mM NaCl) salt stress during its entire growth period. As shown in **Supplementary Fig. 16c**, Nip does not survive under 0.5% salt stress.

According to your suggestions, we have revised **Figure 1C**. We have added a detailed description of the salt resistance treatment process in the figure legend and the Methods section. Scale bars were added in **Figure 1B and 1C**.

2. Related references need to be cited in line 229-231 and line 242-243.

We sincerely appreciate the valuable comments. References were added in the corresponding positions in the revised manuscript.

3. Whether LOC_Os02g08540 and LOC_Os07g35680 under selective sweep during the domestication of *O. rufipogon* to cultivated rice?

Very good question!

We don't think that *LOC_Os02g08540* and *LOC_Os07g35680* underwent selective sweep during the domestication of *O. rufipogon* to cultivated rice. These two wild rice allelic genes were not selected during domestication. Some favorable resistance loci in wild relatives may be linked with adverse agronomic traits. We also checked these two genes according to the results from Huang et

al. (Huang et al. *Nature*, 2012, 490, 497-501), which confirmed that they were not under selective sweep.

4. Does the alleles of LOC_Os02g08540 and LOC_Os07g35680 from Y476 exist in other rice cultivars and accessions? In addition, phylogenetic analysis of LOC_Os02g08540 is suggested to perform.

Thank you for this good suggestion.

The haplotype of the wild rice *LOC_Os02g08540* allele was found in other rice cultivars. We have not found the wild rice haplotype of *LOC_Os07g35680* in cultivated rice, and the same SV has not been found. A phylogenetic tree of *LOC_Os02g08540* was added to the revised manuscript.

5. In Supplemental Fig 14, the grains of N214 and Nip need to be shown. In addition, the panels need to be labeled by ABC.

We sincerely thank you for your valuable feedback regarding our manuscript. The grains of N214 are shown in the revised version, and the panels are labeled.

6. In Supplemental Fig 16, given that the 1,000-grain weight, grain length and grain number per plant of N133 are higher than that of Nip, how about the yield per plant of Nip and N133?

Great question!

We investigated the yield per plant of N133 and Nip, and we added these results to the manuscript with a corresponding Supplementary Figure (S-Fig.18). The yield of N133 was significantly higher than that of Nip under normal conditions. Thank you for your helpful suggestion.

7. A title need to be added in Supplemental Fig 17. The scale bar need to be added in Supplemental Fig 17B.

We sincerely thank you for your careful reading. A title and scale bars were added to the revised corresponding Figure (S-Fig 20). Thank you.

8. The characters of gene names in Fig 6G, 7C-7E, Supplemental Fig 14, Supplemental Fig 15D-15G, Supplemental Fig 17C-17D and Supplemental Fig 20 should be italic.

We are very sorry for our careless mistakes. Thank you for your reminder. We have revised this part of the manuscript. Thank you very much!

Reviewers' Comments:

Reviewer #1:

Remarks to the Author:

First of all, a huge congrats on generating the two haplotype assemblies of Y476. In overall, I find the manuscript has substantially improved in its language and representation, and addressed some of my concerns before. I wish the authors had utilized the two nearly complete haploid assemblies in finding their PAVs and take the full advantage of using a wild, highly heterozygous genome.

Below are my point-by-point comments following the numbers in my previous review.

- 1-1. I was pleased to see the two haplotype assemblies were made, as the authors show in Supplementary Fig. 5, which look nearly complete. I had difficulties finding the assemblies though. The assemblies are only available to download via Figshare, through the link in response to Reviewer #2. The Figshare link in reporting summary is outdated. Besides, only the sequencing reads were available on the Chinese NCB GSA. The “Genome assembly and annotation” folders also contained Illumina reads from the whole genome or transcriptomes. NCBI Bioproject PRJNA936424, which is stated in Nature’s reporting summary for NCBI, is also empty with no public datasets available.
- 1-2. What is the rational for using the primary assembly throughout the manuscript, with missing half the genome? Being ‘gapless’ could be impressive, however biologically, does not provide much insight especially when the interest is on the wide variety of genomic diversity found in a ‘wild’ rice. I would rather prefer using a near-T2T diploid assembly, with a few gaps but with better representation of the two haplotypes for any genomic segment (PAV) studies. It will be also useful to the authors to discover additional PAVs that are not present in the Y476 primary assembly. See the Merqury spectra-asm plots, which show the genomic portion of the 1-copy region is much better covered by the diploid assembly than the primary gapless assembly (Left, diploid hap1 + hap2. Right, haploid gapless assembly). I’d highly encourage to use the diploid assembly throughout the paper, and evaluate with VerityMap and T2T-Polish. I’m sure the amount of errors reported due to haplotype differences will be substantially reduced.

- 1-3. Was polishing applied? In the response, the authors claim “Adhering to your suggestion, we refrained from over-polishing”. No polishing certainly refrains over-polishing, however also means there are still lots of errors left. Since both haplotypes are nearly complete, I was expecting both haplotypes to be polished, using a mapping-based

approach, by using alignments from long and short reads mapped to both haplotype assemblies at the same time.

- 1-4. Which k-mer size was used? Jellyfish parameters in the methods indicate 21-mers. Supp. Fig 2 legend says 17-mers. Methods section states 19-mers being used for Merqury. I performed Merqury with 21-mers, which I think is reasonable given the quality of the reads and assembly.
- 1-5. In table 1, it is not clear to me what IRGC106162 and Yuanjiang wild rice of China is without looking into the references. A little more description about the strain would be helpful. Contig NG50 would be also much more helpful as N50 could be biased. In addition, please add 9311 and Nip assembly summary statistics, as they are frequently compared against Y476 (related to point 13).
- 1-6. The Methods section is missing the diploid assembly part. Also add the other SV validation methods performed in response to Reviewer #2.

2-1. Response Fig. 1 shows a lot of reads with big discrepancy to the assembly, which looks like belong to the other haplotype. The alignment will become much cleaner and convincing if it was shown on the two haplotype assemblies for the same region, using both haplotype as the reference for alignment at the same time. The current Response Fig. 1 indicates the region has large differences between the two haplotypes. How does the corresponding region look like on the two haplotype assemblies? Perhaps a dotplot or ribbon plot showing the SVs between the two haplotypes of the gap region might give a better understanding why it was left as a gap in previous assemblies.

2-2. The authors show validation of the gap regions with PCR in Supp. Fig 8C. I am not a big fan of validating sequences with PCR because it only provides a very small fraction of the replacing sequence, which could be also present elsewhere in the genome. 3-5 kb sounds more suitably alignable with HiFi or ONT reads to show support to the relevant region in the new assembly (or, the two haplotype assemblies). Moreover, if it was left as a gap, the sequences are either too repetitive, or highly heterozygous, resulting in breaks in previous assemblies, which could be also challenging for PCR validation.

3-1. Please add inversion breakpoints to Supp. Fig 7 A-B.

3-2. Please show the ONT read alignments to the corresponding region of Y476, on the two haplotype assemblies. Insure the reads shown in the Nipponbare genome alignment are identical to the ones in Y476 alignment.

4. Please add the results to the main text. I expect the wide range of coverage depth to go smoother when using the alignments to both haplotypes.

5. What's the status of the telomeres on the H1 and H2 assemblies?

6~8. Provide the estimated copy number in Response Table 3, and add it to the main text / Supp. tables.

9. Awesome, please add the validation results to the main text, including observations whether it was present on hap1 or in both haplotypes.

10. The authors present the base level composition difference in the promoter of *LOC_Os02g08540* in Response Table 4. Were there no base level differences, such as SNVs, in the gene (exon) itself? If so, state it also in the main text. Also for *LOC_Os07g35680*.

11. Great!

12. Response Fig. 3 PCA plot clearly shows Y476 is towards the center of Or-I (green dots). Or-II has some overlaps with Or-I, Y476 is outside of the boarder. Given the new PCA plot, I am even more not convinced with the conclusion that Y476 should be classified as Or-II. The PCA and Admixture in Supp. Fig 1 also shows that the classification of Or-I, Or-II, and Or-III is not clear. All I see is the classification being very arbitrary made. In the admixture plot, Or-I looks like a mix of two groups. Y476 has K1, K4, K6 components that is also shared with Or-I. K-5 component could be considered the major genomic component of Or-II, however is also shared in Or-III. The authors should provide a clarification that the Or-I, Or-II, and Or-III was assigned based on Huang et al. in the main text, and discuss this problem. To me, reading through Huang et al.'s paper, it looks like an over-simplified classification of the Ors. Also it is hard to imagine an accurate classification with only 8 M SNPs on a poor reference, using 2x WGS from >1% diverged genomes.

13. See comment on 1-5. Also 1-2. I am sure the authors will be able to discover more valuable PAVs using the hap1 and hap2 assemblies. Would be also useful to the community if you could catalogue the SVs on Hap1 and Hap2 coordinates.

14. Great!

15. Looking forward to seeing your future studies.

16-18. The manuscript reads much better. All my minor comments have been addressed.

REVIEWER COMMENTS

Reviewer #1 (Remarks to the Author):

First of all, a huge congrats on generating the two haplotype assemblies of Y476. In overall, I find the manuscript has substantially improved in its language and representation, and addressed some of my concerns before. I wish the authors had utilized the two nearly complete haploid assemblies in finding their PAVs and take the full advantage of using a wild, highly heterozygous genome.

Below are my point-by-point comments following the numbers in my previous review.

1-1. I was pleased to see the two haplotype assemblies were made, as the authors show in Supplementary Fig. 5, which look nearly complete. I had difficulties finding the assemblies though. The assemblies are only available to download via Figshare, through the link in response to Reviewer #2. The Figshare link in reporting summary is outdated.

Besides, only the sequencing reads were available on the Chinese NCB GSA. The “Genome assembly and annotation” folders also contained Illumina reads from the whole genome or transcriptomes. NCBI Bioproject PRJNA936424, which is stated in Nature’s reporting summary for NCBI, is also empty with no public datasets available.

Thank you for your valuable feedback regarding the accessibility of the NGDC database and the deposition of assemblies.

We have ensured that the assemblies associated with our study are deposited in the National Center for Biotechnology Information (NCBI) and National Genomics Data Center (NGDC). The accession numbers are PRJNA1029807 and PRJCA015108 (<https://ngdc.cncb.ac.cn/bioproject/browse/PRJCA015108>), respectively. This will allow readers to access the assemblies directly from the international repository, providing a reliable and accessible source of the data. The outdated Figshare link in the reporting summary will be updated with the correct link (https://figshare.com/articles/dataset/Genome_sequence_and_annotation_of_Hap1_Hap2_and_primary_of_Y476/24798696) that provides access to the two haplotype assemblies.

We understand the importance of easy and transparent access to data for reproducibility and further research. We deeply regret the oversight and are taking corrective measures to ensure all datasets and assemblies related to our manuscript are easily accessible to the scientific community. Thank you once again for pointing out these discrepancies, and we are committed to rectifying them promptly.

1-2. What is the rationale for using the primary assembly throughout the manuscript, with missing half the genome? Being ‘gapless’ could be impressive, however biologically, does not provide much insight especially when the interest is on the wide variety of genomic diversity found in a ‘wild’ rice. I would rather prefer using a near-T2T diploid assembly, with a few gaps but with better representation of the two haplotypes for any genomic segment (PAV) studies. It will be also useful to the authors to discover additional PAVs that are not present in the Y476 primary assembly. See the Merqury spectra-asm plots, which show the genomic portion of the 1-copy region is much better covered by the diploid assembly than the primary gapless assembly (Left, diploid hap1 + hap2.

Right, haploid gapless assembly). I'd highly encourage to use the diploid assembly throughout the paper, and evaluate with VerityMap and T2T-Polish. I'm sure the amount of errors reported due to haplotype differences will be substantially reduced.

(See pdf attachment for the spectrum plots)

Thank you for your insightful suggestion regarding the use of a diploid assembly in our manuscript. We greatly appreciate your detailed rationale emphasizing the importance of capturing the genomic diversity inherent in wild rice, especially for studies focusing on PAV.

In response to your feedback, we have revised our manuscript to incorporate analyses using a near-T2T diploid assembly. This change allows for a more comprehensive representation of the two haplotypes for any genomic segment, addressing the limitations of the previously used primary gapless assembly. Furthermore, we have conducted genome evaluations using VerityMap (Mikheenko et al., 2020, *Bioinformatics* 36, i75-i83) and T2T-Polish (Mc Cartney et al., 2022, *Nature methods* 19, 687-695), as you recommended. The evaluation results on the Hap1 genome show that the VerityMap results reveal a combined length of possible heterozygous sites and errors amounting to 13.74 Mb, while the low-quality areas identified by T2T-Polish total 4.30 Mb. On the Hap2 genome, the evaluation results indicate that the VerityMap results reveal a combined length of possible heterozygous sites and errors amounting to 14.83 Mb, while the low-quality areas identified by T2T-Polish total 4.18 Mb.

We believe that these revisions significantly improve the manuscript by providing a more accurate and biologically relevant insight into the genomic diversity of wild rice. The manuscript now includes a comparison between the diploid assemblies. We are confident that these changes have addressed your concerns effectively and have enriched the quality and scientific rigor of our work. Thank you once again for your valuable feedback.

References:

Mikheenko, A. et al. TandemTools: mapping long reads and assessing/improving assembly quality in extra-long tandem repeats. *Bioinformatics* 36, i75-i83 (2020).

Mc Cartney, A. M. et al. Chasing perfection: validation and polishing strategies for telomere-to-telomere genome assemblies. *Nat. methods* 19, 687-695 (2022).

1-3. Was polishing applied? In the response, the authors claim "Adhering to your suggestion, we refrained from over-polishing". No polishing certainly refrains over-polishing, however also means there are still lots of errors left. Since both haplotypes are nearly complete, I was expecting both haplotypes to be polished, using a mapping-based approach, by using alignments from long and short reads mapped to both haplotype assemblies at the same time.

Thank you for raising this concern. Indeed, while HiFi reads have a high accuracy, we recognize the potential for errors in our assembly. In light of your suggestion, we revisited the polishing step. We polished both haplotype assemblies using both short reads and HiFi reads with the help of Racon and Merfin through two iterative rounds (Mc Cartney et al., 2022, *Nature methods* 19, 687-695). After polishing, the primary genome's QV value improved from 51.89 to 53.06, the Hap1 genome's QV from 52.39 to 53.58 and the Hap2 genome's QV from 51.31 to 52.71. We appreciate your valuable feedback and have amended our manuscript accordingly to reflect these changes.

References:

Mc Cartney, A. M. et al. Chasing perfection: validation and polishing strategies for telomere-to-telomere genome assemblies. *Nat. methods* **19**, 687-695 (2022).

1-4. Which k-mer size was used? Jellyfish parameters in the methods indicate 21-mers. Supp. Fig 2 legend says 17-mers. Methods section states 19-mers being used for Merqury. I performed Merqury with 21-mers, which I think is reasonable given the quality of the reads and assembly.

Thank you for pointing out the inconsistency.

We used 21-mers for the genome survey and genome quality assessment, as well as in all analyses where the kmer parameter was involved. Based on your recommendation, we have re-conducted the genome survey and genome quality assessment with 21-mers and updated the associated figures in the revised manuscript. The Methods section has also been amended to reflect this change.

1-5. In table 1, it is not clear to me what IRGC106162 and Yuanjiang wild rice of China is without looking into the references. A little more description about the strain would be helpful. Contig NG50 would be also much more helpful as N50 could be biased. In addition, please add 9311 and Nip assembly summary statistics, as they are frequently compared against Y476 (related to point 13).

We genuinely appreciate your constructive feedback regarding the clarity of our manuscript. Adhering to your suggestion, we have added more description and data in Table 1. First, detailed description of IRGC106162 and Yuanjiang wild rice of China are provided in revised Table 1. In addition, we also have added NG50 of Y476 and summary statistics of 9311 and Nip assembly.

1-6. The Methods section is missing the diploid assembly part. Also add the other SV validation methods performed in response to Reviewer #2.

Thank you for highlighting the omission.

We have now included the diploid genome assembly process in the Methods section. Briefly, we used hifiasm to assemble the sequencing data from HiFi, HiC, and UL-reads (hifiasm -o Y476.asm --h1 Y476_HiC_1.fq.gz --h2 Y476_HiC_2.fq.gz --ul Y476_UL.fq.gz Y476.ccs.hifi.fq.gz). We then applied the same scaffolding pipeline to the outputs of primary, hic.hap1, and hic.hap2 assemblies. This resulted in the formation of the diploid genome assembly and a hybrid primary genome.

In addition, we added the SV validation method in the Methods section. The details are as follows:
SV validation methods

(1) To align the HiFi and ONT data of Y476 with the Y476 and Nip genomes, minimap2 (Li et al., 2021, *Bioinformatics* 37, 4572-4574) was employed by utilizing the parameters -ax map-hifi and -ax map-ont, respectively. Subsequently, the aligned data were visualized using the Integrative Genomics Viewer (IGV) (Robinson et al., 2011, *Nature Biotechnology* 29, 24 - 26). Additionally, Y476's Hi-C data were aligned to the Y476 and Nip genomes using HiC-Pro (Servant et al., 2015, *genome biology* 16, 259), and the interaction matrices were visualized using HiCPlotter (Akdemir et al., 2015, *genome biology* 16, 198).

(2) The two ends of SV were amplified by PCR, and the primers were designed based on the two ends of insertion sequence and two genomic common regions adjacent to the insertion ends. Then the DNA of Nip and Y476 were then used as templates for amplification with the designed primers, and the amplified products were subsequently detected by agarose gel electrophoresis.

References:

Li, H. New strategies to improve minimap2 alignment accuracy. *Bioinformatics* **37**, 4572-4574 (2021).

Robinson, J. T. et al. Integrative genomics viewer. *Nat. Biotechnol.* **29**, 24 – 26 (2011).

Servant, N. et al. HiC-Pro: an optimized and flexible pipeline for Hi-C data processing. *Genome Biol.* **16**, 259 (2015).

Akdemir, K. C. & Chin, L. HiCPlotter integrates genomic data with interaction matrices. *Genome Biol.* **16**, 198 (2015).

2-1. Response Fig. 1 shows a lot of reads with big discrepancy to the assembly, which looks like belong to the other haplotype. The alignment will become much cleaner and convincing if it was shown on the two haplotype assemblies for the same region, using both haplotype as the reference for alignment at the same time. The current Response Fig. 1 indicates the region has large differences between the two haplotypes. How does the corresponding region look like on the two haplotype assemblies? Perhaps a dotplot or ribbon plot showing the SVs between the two haplotypes of the gap region might give a better understanding why it was left as a gap in previous assemblies.

We believe that employing both haplotypes for analysis, as you recommended, indeed clarifies the situation and enhances the credibility of our results. We appreciated your guidance in enhancing the clarity of our presentation.

In response to your suggestion, we've taken the approach to use both haplotypes as references and have re-aligned the reads for the corresponding region. With this method, the alignments for HiFi and ONT in both haplotype assembly results appear much cleaner and provide a more accurate representation.

Thank you for your insightful observation regarding Response Fig. 1. The discrepancies observed arise due to the differences between the two haplotypes.

Response Fig. 1. IGV depiction of HiFi and UL reads mapping to the Hap1/Hap2 genome across three gap regions.

(A-F) Above is the coverage depth and read mapping situation of HiFi data at gap upstream and downstream positions; below is the coverage depth and read mapping situation of UL reads. **A, C, E** represent the three gap regions of Chr.5, Chr.8, Chr.12 of Hap1, respectively. **B, D, F** represent the three gap regions of Chr.5, Chr.8, Chr.12 of Hap2, respectively.

2-2. The authors show validation of the gap regions with PCR in Supp. Fig 8C. I am not a big fan of validating sequences with PCR because it only provides a very small fraction of the replacing sequence, which could be also present elsewhere in the genome. 3-5 kb sounds more suitably alignable with HiFi or ONT reads to show support to the relevant region in the new assembly (or, the two haplotype assemblies). Moreover, if it was left as a gap, the sequences are either too repetitive, or highly heterozygous, resulting in breaks in previous assemblies, which could be also challenging for PCR validation.

Thank you for your thorough review and valuable comments on our manuscript. We fully understand your concerns regarding the use of PCR for sequence validation, especially given the potential for sequences to be present elsewhere in the genome.

In response to your suggestion to provide support for the validation regions using HiFi or ONT, we sincerely appreciate your insight and have made corresponding adjustments to our work.

We have now included alignment results for the validation regions of Chr1, Chr9, and Chr11 using HiFi/ONT, employing the two haplotype assemblies for the alignment. To provide a more comprehensive view and facilitate easier comparison, we have integrated both the HiFi/ONT alignment results and the PCR validation results in FigS8 (the revised manuscript is FigS6).

We believe that these revisions will enhance the rigor and clarity of our work. Your feedback has been instrumental in improving the quality of our paper.

3-1. Please add inversion breakpoints to Supp. Fig 7 A-B.

Thank you for the suggestion. We added the inversion breakpoints to Supplementary Figure 7 A-B (Supplementary Figure 5 A-B in revised manuscript) . These breakpoints are now clearly indicated on the figure for better understanding and clarity. We believe this addition will enhance the comprehension of our findings.

3-2. Please show the ONT read alignments to the corresponding region of Y476, on the two haplotype assemblies. Insure the reads shown in the Nipponbare genome alignment are identical to the ones in Y476 alignment.

Thank you for highlighting the oversight in Response Figure 2.

In response, we extracted the reads aligned to the region highlighted in the red box in Response Figure 2 from the bam file. Subsequently, we realigned these reads to both haplotype assembly

genomes of Y476. Our results demonstrate that the ONT sequencing data aligns consistently with the two haplotype assemblies of Y476.

We appreciate your patience and guidance in ensuring the comprehensive presentation of our findings.

Response Fig. 2. UL read maps span across the SV regions.

(A-J) the coverage depth and read mapping situation of UL reads at SV regions. **A, C, E, G, I** represent the five SVs regions of Chr.2, Chr.6, Chr.7, Chr.9 and Chr.12 of Hap1, respectively. **B, D, F, H, J** represent the five SVs regions of Chr.2, Chr.6, Chr.7, Chr.9 and Chr.12 of Hap2, respectively.

4. Please add the results to the main text. I expect the wide range of coverage depth to go smoother when using the alignments to both haplotypes.

Thank you for pointing this out. We have incorporated the results into the main text as you suggested. In addition, we have also added the relevant results in the Supplementary Table 5. Upon using the alignments to both haplotypes, we observed a more uniform coverage depth across the genome. This observation confirms your expectations and underscores the value of considering haplotype resolved in assembly. We appreciate your valuable feedback which has led to a clearer presentation of our findings.

5. What's the status of the telomeres on the H1 and H2 assemblies?

Thank you for your inquiry. Indeed, in Table 1 we have described the number of telomeres in the H1 and H2 assemblies. However, we recognize that there was a lack of detailed description. We have now included more comprehensive details in Supplementary Table 10. In essence, both of the H1 and H2 assemblies are consistent with the primary genome with 20 telomeres each, missing the left telomeres for chromosomes 9 and 10. We have added the detailed count and length information in the supplementary Table 10.

6~8. Provide the estimated copy number in Response Table 3, and add it to the main text / Supp. tables.

Thank you for your suggestion to provide estimated copy numbers.

To address this, we utilized minimap2 to map these genes back to both haplotype genomes. By combining this with the sequencing coverage, we have estimated the copy number for these genes. This updated information has now been incorporated into the supplementary table 15 for clarity and completeness. Furthermore, we've included the coverage and copy number data for *LOC_Os04g32350* and *LOC_Os11g44960* (NBS-LRR), as well as *LOC_Os11g45050* (NBS-LRR) and *LOC_Os11g45090* (NBS-LRR).

We believe these additions provide a more comprehensive view of our findings and appreciate your guidance in enhancing the depth of our presentation.

9. Awesome, please add the validation results to the main text, including observations whether it was present on hap1 or in both haplotypes.

We sincerely appreciate your suggestions. We amplified the CDS region and promoter region of the *LOC_Os07g35680* and *LOC_Os02g08540* of Y476 by PCR and sequenced the amplified products, the result of sequencing was consistent with the assembly sequence of Y476 genome. Based on the assembled Y476 haplotype genome, the difference between two haplotype sequences of *LOC_Os07g35680* gene and Nip sequence was compared, the comparison results showed that there was a 7.8 kb SV between Hap2 and Nip, while it was missing between Hap1 and Nip. For the *LOC_Os02g08540*, the 87-bp SV between Y476 and Nip is present in Hap1 and Hap2 genomes. And the results were verified by PCR amplification and sequencing. Adhering to your suggestion, we added the validation results to the main text.

10. The authors present the base level composition difference in the promoter of *LOC_Os02g08540* in Response Table 4. Were there no base level differences, such as SNVs, in the gene (exon) itself? If so, state it also in the main text. Also for *LOC_Os07g35680*.

We greatly appreciate your queries. there were base level differences in the gene (exon) itself between Nip and Y476. For *LOC_Os02g08540*, there were three non-synonymous mutations in the coding region between Nip and N133. For *LOC_Os07g35680*, there were several non-synonymous mutations in the coding region between Nip and N154. Thank you for your advice. According to your suggestions, we have added the relevant description in the main text and add the variation of CDS region in the supplementary table 25 and supplementary table 28, respectively.

11. Great!

Thank you again for your valuable feedback earlier.

12. Response Fig. 3 PCA plot clearly shows Y476 is towards the center of Or-I (green dots). Or-II has some overlaps with Or-I, Y476 is outside of the boarder. Given the new PCA plot, I am even more not convinced with the conclusion that Y476 should be classified as Or-II. The PCA and Admixture in Supp. Fig 1 also shows that the classification of Or-I, Or-II, and Or-III is not clear. All I see is the classification being very arbitrary made. In the admixture plot, Or-I looks like a mix of two groups. Y476 has K1, K4, K6 components that is also shared with Or-I. K-5 component could be considered the major genomic component of Or-II, however is also shared in Or-III. The authors should provide a clarification that the Or-I, Or-II, and Or-III was assigned based on Huang et al. in the main text, and discuss this problem. To me, reading through Huang et al.'s paper, it looks like an over-simplified classification of the Ors. Also it is hard to imagine an accurate classification with only 8 M SNPs on a poor reference, using 2x WGS from >1% diverged genomes.

We very appreciated your insightful comments. From the PCA, admixture, and phylogenetic analysis in our manuscript, we realized that the conclusion that “Y476 should be classified as Or-II” is arbitrary. Our study on the subpopulation identification of Y476 was carried out on the basis of Huang et al.'s study, which divided 446 wild rice into three subpopulations, Or-I, Or-II, Or-III (Huang et al., 2012, *Nature* 490, 497-501). We totally agree with you that it like an over-simplified classification of the Ors with twofold genome coverage. According to your comments, we revised the corresponding part in the main text and the conclusion was revised as that Y476 was among Or-clusters, but not clearly classified into a specific subgroup. In addition, the phylogenetic analysis in Fig. 2A in previous manuscript was moved to Supplementary Fig. 1 in the revised manuscript. Moreover, we also think that a larger population of wild rice species and higher quality sequencing data may be more conducive to the clear division of Y476 subpopulation and the Y476 genome can serve as an important reference genome for the next step of wild rice classification and data analysis. Moreover, we have added a discussion on phylogenetic analysis in the discussion section.

References:

Huang, X. et al. A map of rice genome variation reveals the origin of cultivated rice. *Nature* **490**, 497-501 (2012).

13. See comment on 1-5. Also 1-2. I am sure the authors will be able to discover more valuable PAVs using the hap1 and hap2 assemblies. Would be also useful to the community if you could catalogue the SVs on Hap1 and Hap2 coordinates.

Thank you for your insightful feedback. In response to your comments, we have identified SVs by comparing the polished Y476 Hap1/Hap2 genomes with T2T-Nip. The coordinates and sizes of the SVs have been added to Supplementary Table 11. We believe that cataloguing these SVs based on the coordinates of Hap1 and Hap2 will be of great value to the community, and we appreciate your suggestion in this regard.

14. Great!

Thank you for your feedback.

15. Looking forward to seeing your future studies.

Thank you again for pointing us in a new direction for future research.

16-18. The manuscript reads much better. All my minor comments have been addressed.

It is on the basis of your and other reviewers of the previous valuable comments that our manuscript read much better, thank you again.

In conclusion, we are committed to further refining our manuscript based on your invaluable feedback. Your insights are pivotal in enhancing the quality and depth of our research. We hope that our responses address your concerns and we are open to any further suggestions you might have.

Reviewer #2 (Remarks to the Author):

The authors have addressed some of my concerns, in particular they have made the data available in a public archive and have updated the assembly methods. However, there are still several aspects of the analysis that need to be clarified and/or improved.

- The figures and tables added as part of the author response should be included in the manuscript somewhere in high resolution. As provided now, I had trouble reading the screenshots of IGV as zooming in made the text too blurry to read.

We sincerely apologize for the previous ambiguity caused by the low resolution of some figures and tables. Thank you for your thoughtful suggestion. We have improved the resolution of the pictures and tables in the revised manuscript.

- The assembly strategy remains unclear. Supplementary Figure 6 is not updated to show the new strategy. The manuscript seems to mention multiple hifiasm runs with different parameters some of which are not valid options (e.g. -hic is not a valid option). It is also not clear which assembly was used for the analysis. The authors seem to have generated 3 assemblies, 1 w/o HiC data and two haplotypes with Hi-C phasing. However, it seems the first assembly was used for all analysis. If this is the case, it would not be haplotype resolved and would have similar issues with haplotype switches as the previous version and would lead to ambiguous read mappings from diverged reads from the second haplotype.

Thank you for pointing out the ambiguities in our assembly strategy and the need for clarity in our manuscript. We recognize the importance of a clear and precise description of our genomic assembly process, especially in light of your concerns regarding the potential for haplotype switches and ambiguous read mappings.

In response to your feedback, we have thoroughly revised the relevant sections of the manuscript. The updated manuscript now clearly outlines our revised assembly strategy. Specifically, we have used Hifiasm with the parameters --h1 --h2 -ul, incorporating HiFi, UL, and Hi-C data. This approach allowed us to generate a more accurate and comprehensive diploid assembly.

To clarify the utilization of different assemblies in our analysis, the revised manuscript states that the diploid assembly was used for subsequent analyses and evaluations. This addresses the issue of potential haplotype switches and ambiguous read mappings. This change allows for a more comprehensive representation of the two haplotypes for any genomic segment, addressing the limitations of the previously used primary gapless assembly.

We believe these revisions provide a clearer understanding of our assembly approach and its implications on the study's outcomes. The manuscript now includes a comparison between the diploid assemblies. We appreciate your guidance in this matter and hope that these changes meet your expectations for clarity and precision in our methodology.

- It is also not clear what role Hi-C scaffolding played. The authors state they got 12 contigs without gaps but then these were oriented and ordered using HiC. If the 12 contigs are chromosomes what would Hi-C be orienting?

Thank you for your perceptive observation and suggestion regarding Hi-C data.

While our primary genome assembly resulted in 12 contigs that align with the expected 12 chromosomes of the rice genome, the importance of Hi-C comes in validating the assembly's accuracy. By using Hi-C data, we were able to compute interaction intensities and subsequently visualize these intensities in a heatmap. This heatmap allowed us to ensure that our assembled genome followed the expected patterns of chromosomal interactions.

Furthermore, our hap1 and hap2 assemblies initially resulted in 15 contigs each. The use of Hi-C data here was crucial to correctly connect and order these contigs to arrive at the expected 12 chromosomes. The resulting Hi-C heatmap further provided validation of these assemblies.

To provide a clearer understanding and address the concerns raised by you and other reviewers, we have revised our Methods section. This revision provides a detailed description of our assembly process for both the primary genome and the diploid genome. We believe this added detail will

clarify the pivotal role of Hi-C in our assembly and validation processes. Once again, thank you for bringing this to our attention.

- Was the validation performed on only the primary assembly or on both haplotype assemblies?

We greatly appreciate your queries. the validation performed on both haplotype assemblies.

- Response Figure 1 is not convincing for correctness of the assembly. The coverage still fluctuates significantly and there are variants supported by many reads, in some cases 50/50 but others appear to be almost all the reads. If this is all reads vs a primary assembly, some of the variants and coverage fluctuations may be due to the other haplotype reads. Using the two haplotypes for analysis instead as I suggested above would help clarify this.

We believe that employing both haplotypes for analysis, as you recommended, indeed clarifies the situation and enhances the credibility of our results. We appreciated your guidance in enhancing the clarity of our presentation.

In response to your suggestion, we've taken the approach to use both haplotypes as references and have re-aligned the reads for the corresponding region. With this method, the alignments for HiFi and ONT in both haplotype assembly results appear much cleaner and provide a more accurate representation.

Thank you for your insightful observation regarding Response Fig. 1. The discrepancies observed arise due to the differences between the two haplotypes.

Response Fig. 1. IGV depiction of HiFi and UL reads mapping to the Hap1/Hap2 genome across three gap regions.

(A-F) Above is the coverage depth and read mapping situation of HiFi data at gap upstream and downstream positions; below is the coverage depth and read mapping situation of UL reads. **A, C, E** represent the three gap regions of Chr.5, Chr.8, Chr.12 of Hap1, respectively. **B, D, F** represent the three gap regions of Chr.5, Chr.8, Chr.12 of Hap2, respectively.

- Response figure 2 doesn't seem to address the reviewer's question. While it shows the reads do not align well to the NIP assembly, how do the same reads align to the Y476 assembly the authors are using?

Thank you for highlighting the oversight in Response Figure 2.

In response, we extracted the reads aligned to the region highlighted in the red box in Response Figure 2 from the bam file. Subsequently, we realigned these reads to both haplotype assembly genomes of Y476. Our results demonstrate that the ONT sequencing data aligns consistently with the two haplotype assemblies of Y476.

We appreciate your patience and guidance in ensuring the comprehensive presentation of our findings.

Response Fig. 2. UL read maps span across the SV regions.

(A-J) the coverage depth and read mapping situation of UL reads at SV regions. **A, C, E, G, I** represent the five SVs regions of Chr.2, Chr.6, Chr.7, Chr.9 and Chr.12 of Hap1, respectively. **B, D, F, H, J** represent the five SVs regions of Chr.2, Chr.6, Chr.7, Chr.9 and Chr.12 of Hap2, respectively.

- I wasn't clear what the takeaway of response table 3 should be. The genes show a wide range of coverage, from close to the mean coverage to more than 3 SD away. This table should include some expectation of copy number from the assembly to see if the coverage matches expectation. Also, the gene NBS-LRR should be included since the reviewer explicitly asked about its copy number and I did not see coverage results in Figure 3.

Thank you for your feedback, and we apologize for any confusion regarding Response Table 3.

To clarify, we utilized miniprot to map these genes back to both haplotype genomes. By combining this with sequencing coverage, we estimated the copy number for these genes. We have now incorporated this information into the Supplementary Table 15 to provide a clearer picture of how the observed coverage correlates with the expected copy number based on our assembly.

Furthermore, in line with your suggestion, we've added coverage and estimated copy number data for the NBS-LRR genes. Specifically, we've included data for *LOC_Os04g32350* and *LOC_Os11g44960* (NBS-LRR), as well as *LOC_Os11g45050* (NBS-LRR) and *LOC_Os11g45090* (NBS-LRR). Notably, the gene *LOC_Os04g32350* exhibited a copy number variation spanning the entire gene, corresponding with the genes cluster of Y476.Chr04.657- Y476.Chr04.718. The three NBS-LRR genes from Nip underwent domain expansion, corresponding to Y476.Chr11.2499- Y476.Chr11.2573, which represents a gene cluster in Y476 containing the NBS-LRR gene domain. We hope these additions and clarifications better address your queries and elucidate the insights derived from our data.

- The citation to hifiasm w/UL support should be: <https://arxiv.org/abs/2306.03399>. The authors refer to the assembly method as "novel" but I'm not clear what this means as it is a previously published assembly method from an unrelated set of authors. Does novel here mean novel to the rice community?

Thank you for pointing out the citation error. We have now corrected the reference for hifiasm. When we referred to the method as 'novel', we intended to convey that it is a recently released method (approximately half a year ago on GitHub after we submitted this manuscript) for integrating HiFi and UL data. I apologize for the confusion caused by my choice of words. We have replaced "novel" with "newly developed". Additionally, as far as the rice community is concerned, we are among the first to use this method for a published genome.

- Is there a reason the assembly is only made available through figshare and not the public sequencing database?

Thank you for your valuable feedback regarding the accessibility of assemblies.

We have ensured that the assemblies are deposited in the National Center for Biotechnology Information (NCBI) and National Genomics Data Center (NGDC). The accession numbers are PRJNA1029807 and PRJCA015108 (<https://ngdc.cnpc.ac.cn/bioproject/browse/PRJCA015108>), respectively. This will allow readers to access the assemblies directly from the international repository, providing a reliable and accessible source of the data. The outdated Figshare link in the reporting summary will be updated with the correct link

(https://figshare.com/articles/dataset/Genome_sequence_and_annotation_of_Hap1_Hap2_and_primary_of_Y476/24798696) that provides access to the two haplotype assemblies.

We understand the importance of easy and transparent access to data for reproducibility and further research. We deeply regret the oversight and are taking corrective measures to ensure all datasets and assemblies related to our manuscript are easily accessible to the scientific community.

- There should be a citation to the Huang et al 2012 paper when Or-I, Or-II, etc classes are introduced in the manuscript if this is the origin of the naming and the classification of Y476.

We greatly appreciate your queries. Yes, the classification of Y476 follows the classification criterion of Or-I, Or-II, Or-III proposed by Huang et al. Thanks for your suggestion, In the new manuscript, where Or-I, Or-II, Or-III were introduced, we have cited the 2012 paper by Huang et al.

- The authors clarified in the response that the assembly was oriented with respect to the published rice genome but this is never mentioned in the methods section and should be.

We appreciate your careful review and feedback on the methods. We oriented the assembly scaffolds using Nucmer (Marçais et al., 2018, *PLoS Computational Biology* 14, e1005944) and Ragtag (Alonge et al., 2022, *genome biology* 23, 258) in relation to the published rice genome to ensure that the orientation of the scaffold chains and IDs corresponds one-to-one with the rice chromosome. It's worth noting that this step did not alter any base sequences. We have now added these to the methods section. We appreciate your feedback.

References:

Marçais, G. et al. MUMmer4: a fast and versatile genome alignment system. *PLoS Comput. Biol.* **14**, e1005944 (2018).

Alonge, M. et al. Automated assembly scaffolding using RagTag elevates a new tomato system for high-throughput genome editing. *Genome Biol.* **23**, 258 (2022).

Reviewer #3 (Remarks to the Author):

The manuscript reported a near complete genome of wild rice Y476. Although authors tried great effort to validate the function of structural variations, some major concerns are still not addressed.

i) In line 308-314, the authors reported an insertion in promoter of *SWEET14* in Y476, and this SV reduced the expression of *SWEET14*. In Reference 30, previous study showed that overexpression

of *SWEET14* could increase grain weight. But the knocking-out line of *SWEET14* did not show any change of seed size. In the study “BMC Plant Biol 2020 Jul 3;20(1):313.doi: 10.1186/s12870-020-02524-y”. Other researchers also did not observe any change of seed weight. Moreover, In the study “*OsSWEET14* cooperates with *OsSWEET11* to contribute to grain filling in rice”, the researchers found that the double mutant *ossweet14 ossweet11* has more severe deficiency of grain filling than that of *ossweet11*. Thus, it is hard to conclude that reduced expression of *OsSWEET14* in N124 CSSL line could increase seed size and weight.

We greatly appreciated your insightful comments.

We carefully read two references you mentioned, and Ref. 30 in our previous manuscript (it is Ref. 31 in the revised manuscript), in these paper, *OsSWEET14* negatively regulated rice blight (*Xanthomonas oryzae* pv. *Oryzae*) but not affect the agronomic traits, overexpression of *SWEET14* could REDUCE grain weight. But the knocking-out line of *sweet14* did not show any change of seed size. We deeply agree with you, it is hard to conclude that reduced expression of *OsSWEET14* in CSSL line increase seed size. In view of this point, we deleted the conclusion “different expression of *OsSWEET14* contribute to the grain weight phenotype”. We sincerely apologize for our un-rigorous description.

In this paragraph, we described the relationship between SV and differentially expression gene (DEGs), 82% of the DEGs harbored SVs in their promoter regions, suggesting that SVs play important roles in regulating the expression of these genes. The *OsSWEET14* allele here as an example, we confirmed the expression pattern in N214 (and wild rice Y476) and Nip with qPCR repeatedly, the conclusion “this SV may be the cause of the significant difference in the expression of the *OsSWEET14* gene in Y476 in comparison with Nip” should be correct. The increased grain size in N214 might cause by other introgression segments from wild rice.

ii) In figure 6B and 6C, the author did not label plant Nip and N133, or did not describe it in figure legend. Please add more detailed information of plant.

We thank your valuable feedback. Plants in Figure 6B and 6C were labeled. Thank you very much.

iii) In line 354-Line 403, the authors put great effort and dissect QTL of rice blast resistance and try to identify candidate gene of one QTL. The result found a 7.8-kb insertion in intron of *LOC_Os07g35680* in Y476.

First, the author did not describe the pathogen isolates used in Figure 7.

We genuinely appreciate your constructive feedback regarding the clarity of our manuscript. The plants were inoculated with blast isolates “FJ07-5-2” and “FJ07-8-1”. And we have added descriptions about the pathogen isolates in Figure 7 in revised manuscript.

Second, the author did not label which intron this insertion was located in.

Thank you for highlighting the omission in the manuscript about which intron the insertion was located in. In the revised manuscript, we have revised Figure 7D to address this issue.

Moreover, when a long insertion in intron, the amplification and sequencing of full-length transcript should be confirmed first. If it could not produce full-length transcript in N154/Y476, the resistance may result from new transcript. If it could generate full-length transcript, the expression level can be evaluated. As the author showed, *LOC_Os07g35680* is expressed in Nip in Figure 7C, why Nip is not immune to isolate of *M. oryzae* used in this study.

Great question!

Through amplification and sequencing, full-length transcripts of *LOC_Os07g35680* were generated in N154. There are multiple non-synonymous mutations in coding region between N154 and Nip (we provided this data in revised Supplementary Table 28). The variation of this gene in Nip may contributed to its sensitive to rice blast. In this study, we knockout the *LOC_Os07g35680* in the background of N154, and the knockout lines showed sensitivity to rice blast, suggests that wild rice *LOC_Os07g35680* allele is a functional allele. Given that Nip showed sensitive to rice blast isolates, we did not knock out the *LOC_Os07g35680* allele in the Nip background. Of course, it cannot be ruled out that the gene expression level of *LOC_Os07g35680* plays a cumulative regulatory role on rice blast resistance, rather than have or have not.

Then, I found the qPCR primers “ F-CTGCTGGACTTCGATCACAA , R-GTCAATGGATTGCTCCGAGT” are not specific to *LOC_Os07g35680*, it can also amplify *LOC_Os07g35660* with 100% identity and similar length of PCR product. Thus the author should provide more results to decide whether new transcript or gene expression result in the resistance.

LOC_Os07g35660 and *LOC_Os07g35680* belong to the same gene family, and they have high sequence consistency. We sincerely apologize for our neglect of this issue and wholeheartedly concur with your advice. Thank you again for pointing out this. We redesigned the qPCR primers which specific to *LOC_Os07g35680*, the newly designed primers sequences are as followings and provided in Supplemental Table 30.

F: AGATCTCGGACTTTGGGTTAGCG;

R: CAGATCGTCAGCTGAACGGC.

At the same time, we repeated the qPCR experiments of *LOC_Os07g35680* gene expression using new qPCR primers, the data were updated in the revised manuscript.

Third, when I use target 1 sequence “TGGCGACGGCCATACGACAC” of *LOC_Os07g35680* in figure S24 to search sequence in Nip, it could match to genome sequence of Nip, there should be amino acids variations between Nip and N154/Y476 in locus *LOC_Os07g35680*. It also may be the casual change which resulted in rice blast resistance.

We greatly appreciate your insightful comments. As we respond in last question, there are indeed several nonsynonymous mutations that cause amino acid changes between Nip and N154. The target sequence was designed by N154 *LOC_Os07g35680* allele, not by Nip genome, so it could NOT match to genome sequence of Nip. I think you also found It could not match to Nip genome.

We wholeheartedly concur with your view that amino acids variations may be the casual change which resulted in rice blast resistance. In this study, we focus on the SV which affect this gene's expression level. To confirm whether the amino acids variations in coding region change its function, allele from Nip should be overexpress or knock out in Nip background. Your valuable suggestions have provided directions and ideas for our future studies.

Fouth, in figure S24 the target 1-sequencing map of KO1 and KO4 of *LOC_Os07g35680* clearly showed that one strand is WT, one strand is knock-out type. The target 2 sequencing map of KO1, KO3 and KO4 clearly showed that one strand is WT, one strand is knock-out type. In total, KO1 and KO4 should be heterozygous which one strand is WT. Why the sequential number (KO1, KO2, KO3) in Figure 7 is not consistent with the sequential number in Figure S24? Why KO1 and KO2/4 of *LOC_07gOs35680* is not immune to the isolate used in this study given that one stand of KO1 and KO2/4 harbors the gene *LOC_Os07g35680* from N154/Y476?

Thank you for your valuable feedback. In this study, we used homozygotes of knockout lines for rice blast experiments. The T0 and heterozygous T1 generation of cas9/CRISPR edited plants did not used for blast resistance investigation. Your comment on homozygosity and heterozygosity of the knockout lines in the Figure S24 reminds us that we have provided a wrong Figure S24 that does not match the Figure 7. We deeply apologize for this mistake. We provided the target sequences of the homozygote knockout lines in renewed Figure S24 (it is Figure S22 in the revised manuscript), which consistent with Figure 7.

Fifth, *Pish* may have function in Nip background, *Pi9* should not have function in Nip background. What does the author want to tell if *Pish* did not express in N154 and the expression of *Pi9* has been increased.

According to DEGs from transcriptomes of N154 and Nip after rice blast infection, we selected four rice blast related genes, *Pi9*, *OsWAK112d*, *OsMADS26* and *Pish*. To confirm the regulation pathway *LOC_Os07g35680* involved, we detected the expression levels of these four rice blast resistance genes in the N154-KO lines, which was sensitive to rice blast. Only *OsMADS26* expression was significantly changed in the N154-KO lines, and the change was consistent with that of Nip. So, we made the conclusion "*LOC_Os07g35680* leads to changes in the transcription of rice blast resistance gene *OsMADS26*". Based on the results of expression pattern in N154 knockout line, we don't think *Pish* and *Pi9* are directly regulated genes of *LOC_Os07g35680*.

In brief, it is hard to make me confirm *LOC_Os07g35680* is the causal gene at present stage.

Thank you. When we mentioned *LOC_Os07g35680* in manuscript, we identified it as a "candidate gene" for rice blast resistance. We agree with you that more evidences were needed to confirm it is the causal gene. The homozygotes of *LOC_Os07g35680*-knockout lines in N154 genetic background showed a sensitive phenotype for rice blast, and *OsMADS26* expression was significantly changed in the KO lines. Based on these results, we made the conclusion "the natural variation in the rice *LOC_Os07g35680* locus is critical for resistance to *M. oryzae* and leads to

changes in the transcription of rice blast resistance gene *OsMADS26*".

Reviewer #4 (Remarks to the Author):

The revised manuscript is now substantially improved. Through new experimental data or additional explanation, the authors have responded to all my comments and basically addressed my concerns. This is a impressive study about reference genome and new gene identification of rice. I have no further comments to add.

Thank you very much!

Reviewers' Comments:

Reviewer #1:

Remarks to the Author:

I am pleased to see the overall improvements of this manuscript. I see that most of my comments have been addressed. The manuscript reads much better than before.

Below are a few points that still requires the authors attention.

1. The manuscript is missing “Data Availability” and “Code Availability” section. Assemblies are again, only available on Figshare. I was expecting them to be accessible in public archives, such as NCBI or NGDC.
2. I wanted to bring to the authors’ attention that there are haplotype specific regions lost, which was present in the previous round of revision. The new haplotype resolved assembly is missing roughly 540 kb of haplotype specific sequences compared to the last round. I am not expecting the authors to go back and re do all their analysis, as the conclusion based on the current diploid genome would remain, but it would be useful to at least understand what has been lost for the authors’ future studies.

As visible in the Merqury kmer spectrum (spectra-asm) figure below, the black “read-only” line shows a significant portion around 25x; indicating there are lost genomic segments in the diploid assembly. I have re-attached the same plot from last round to the right. There was almost no missing sequences in the 1-copy region.

3. Please indicate which genome assembly was used as the reference for Supplementary Figs. 5 and 6. Specifically, I assumed Supp Fig 5 AB were showing mapping against Nip, but realized it’s not mentioned in anywhere. Likewise, Supp Fig 6 D-I should label Y476 diploids somewhere for the reference.
4. The newly written portion of the diploid assembly construction and quality assessment are missing citations to all the tools used. I’d suggest to add the missing references, or further simplify the text with no tool names as the newly added Methods section has the proper citations.

Reviewer #2:

Remarks to the Author:

I thank the authors for the comprehensive revisions. The updated manuscript addresses all my major points. I had a few minor remaining questions:

- I could not find the assembly under NCBI. The raw data is available but the assembly is not yet visible?

- The number of VerityMap errors is high, thousands per assembly. Even filtering by length and read support (2000, 25% based on recent publications; Rautiainen et al. 2023, <https://www.nature.com/articles/s41587-023-01662-6>) still leaves close to 1000 errors. This is much higher than reported on a human assembly in Rautiainen et al. 2023. There are no details on how VerityMap was run so I'm not sure if the hap1/hap2 assemblies were provided together to be evaluated as a diploid result or if all reads were used to evaluate each assembly separately. The latter would potentially explain the high rate of errors since there would be many heterozygous differences picked up as "errors".

The manuscript list HiC as scaffolding/orienting the primary assemblies and only listed as validating the Hap1/Hap2 assemblies (e.g. page 6 and page 19). The response text stated scaffolding was only done on Hap1/2 while validation was performed using HiC on all assemblies. The manuscript text should be updated to clarify exactly which assemblies were validated with HiC and which were scaffolded via 3DDNA.

Page 5 according should be according

Page 6 first assembly should be combining HiFi and ONT not HiC and ONT since it's the unphased assembly

Page 7 "nearly 90% coverage of for three data types", 90% of what? I assume this means that 90% of the assembly is covered by all three datatypes with the expected coverage but this is not clear from the current text.

Figure 1 in the review responses shows a likely indel on chr8 hap2. Is this pre-polishing? Was this polished with the authors' method?

Reviewer #3:

Remarks to the Author:

The revised manuscript has addressed all my concerns.

In line 415-416, "adaptive molecular response" is confusing. Although the author may not want to express the response of immune system in that description, I need to say that there is only innate immune system in plant. Please change your description.

Reviewer #1 (Remarks to the Author):

I am pleased to see the overall improvements of this manuscript. I see that most of my comments have been addressed. The manuscript reads much better than before.

Below are a few points that still requires the authors attention.

1. The manuscript is missing “Data Availability” and “Code Availability” section. Assemblies are again, only available on Figshare. I was expecting them to be accessible in public archives; such as NCBI or NGDC.

Response:

Thank you for your valuable feedback regarding the "Data Availability" and "Code Availability" sections of our manuscript. In response to your comment, we have now included detailed "Data Availability" and "Code Availability" sections within our manuscript.

The datasets of this study have been uploaded to both the National Center for Biotechnology Information (NCBI) and the National Genomics Data Center (NGDC). The accession numbers and links to these resources have been clearly stated in the newly added sections.

We believe these updates address your concerns about data and code accessibility and hope that our efforts meet your expectations and the standards of the journal.

2. I wanted to bring to the authors’ attention that there are haplotype specific regions lost, which was present in the previous round of revision. The new haplotype resolved assembly is missing roughly 540 kb of haplotype specific sequences compared to the last round. I am not expecting the authors to go back and re do all their analysis, as the conclusion based on the current diploid genome would remain, but it would be useful to at least understand what has been lost for the authors’ future studies.

As visible in the Merqury kmer spectrum (spectra-asm) figure below (see attached pdf), the black “read-only” line shows a significant portion around 25x; indicating there are lost genomic segments in the diploid assembly. I have re-attached the same plot from last round to the right. There was almost no missing sequences in the 1-copy region.

Response:

We greatly appreciate your detailed examination of our revised haplotype-resolved assembly and your observations regarding the loss of haplotype-specific sequences. Indeed, as indicated by the Merqury kmer spectrum analysis you highlighted, approximately 540 kb of haplotype-specific sequences appear to be missing in the assembly. We are committed to continuous improvement in our research methods and thank you for bringing this matter to our attention.

3. Please indicate which genome assembly was used as the reference for Supplementary Figs. 5 and 6. Specifically, I assumed Supp Fig 5 AB were showing mapping against Nip, but realized it’s not mentioned in anywhere. Likewise, Supp Fig 6 D-I should label Y476 diploids somewhere for the reference.

Response:

Thank you for your attentive reading of our manuscript and for pointing out the need for clearer references to the genome assemblies used in Supplementary Figures 5 and 6. Your observations are crucial for ensuring the clarity and reproducibility of our findings. We have now amended the figure legends and text now clearly indicate this reference, providing the necessary context for interpreting the mapping results.

We believe that these amendments will greatly enhance the clarity and utility of the figures, aiding readers in understanding the genomic contexts and the specific reference genomes used in our analyses.

4. The newly written portion of the diploid assembly construction and quality assessment are missing citations to all the tools used. I'd suggest to add the missing references, or further simplify the text with no tool names as the newly added Methods section has the proper citations.

Response:

Thank you for your constructive feedback on the manuscript. To rectify this, we have thoroughly revised the text to ensure that all tools mentioned are now accompanied by the appropriate references. We hope these revisions satisfactorily address your concerns and improve the manuscript's overall quality and reliability. We are grateful for your attention to detail and guidance in enhancing the scholarly rigor of our work.

Reviewer #2 (Remarks to the Author):

I thank the authors for the comprehensive revisions. The updated manuscript addresses all my major points. I had a few minor remaining questions:

1. I could not find the assembly under NCBI. The raw data is available but the assembly is not yet visible?

Response:

Thank you for your valuable feedback regarding the visibility of the assembly. Previously we only uploaded the assembly in figshare (https://figshare.com/articles/dataset/Genome_sequence_and_annotation_of_Hap1_Hap2_and_primary_of_Y476/24798696). Thank you for your reminder, and now the genome assembly is also available in the National Genomics Data Center (NGDC) under the Bioproject accession number PRJCA015108 (<https://ngdc.cnbc.ac.cn/bioproject/browse/PRJCA015108>). Meanwhile, we have uploaded the assembly data to NCBI, but it is currently awaiting review and has not been released yet.

2. The number of VerityMap errors is high, thousands per assembly. Even filtering by length and read support (2000, 25% based on recent publications; Rautiainen et al. 2023, <https://www.nature.com/articles/s41587-023-01662-6>) still leaves close to 1000 errors. This is much higher than reported on a human assembly in Rautiainen et al. 2023. There are no details on how VerityMap was run so I'm not sure if the hap1/hap2 assemblies were provided together to be

evaluated as a diploid result or if all reads were used to evaluate each assembly separately. The latter would potentially explain the high rate of errors since there would be many heterozygous differences picked up as "errors".

Response:

Thank you for your detailed observation regarding the high number of VerityMap errors reported in our assemblies. Your concern has led us to re-evaluate our approach and methodology with respect to the use of VerityMap. Initially, as you accurately surmised, we utilized all sequencing data to independently assess each genome assembly. This method indeed identified a large number of heterozygous differences as errors, contributing to the unexpectedly high error count.

Taking your valuable feedback into consideration, we have revisited our VerityMap analysis on a merge hap1/hap2 assemblies. This revised analysis has resulted in a significant reduction in the number of detected errors. We have updated our manuscript to reflect these new findings, with the revised error counts now presented in Supplementary Table 8.

We are grateful for your insightful suggestion.

3. The manuscript list HiC as scaffolding/orienting the primary assemblies and only listed as validating the Hap1/Hap2 assemblies (e.g. page 6 and page 19). The response text stated scaffolding was only done on Hap1/2 while validation was performed using HiC on all assemblies. The manuscript text should be updated to clarify exactly which assemblies were validated with HiC and which were scaffolded via 3DDNA.

Response:

Thank you for your insightful query regarding the use of HiC for scaffolding/orienting the primary assemblies and its role in validating the Hap1/Hap2 assemblies as outlined in our manuscript. We acknowledge the confusion caused by our previous descriptions and have taken steps to clarify these processes in our text.

In the revised manuscript, we have eliminated redundant content in the results section and provided a more detailed description in the methods section. To clarify, after filtering, the primary assemblies were reduced to 12 contigs. We employed RagTag to anchor these contigs to the chromosomes of the R498 reference genome. In contrast, the contigs resulting from the Hap1/2 assemblies numbered greater than 12 and were subsequently scaffolded via the 3D-DNA approach. Importantly, all three genomes underwent validation using HiC interaction heatmaps. This step was crucial for verifying the integrity and accuracy of the scaffolded assemblies, ensuring that the chromosomal organization reflects true genomic architecture.

We appreciate the opportunity to clarify these aspects of our methodology and hope that these revisions satisfactorily address your concerns.

4. Page 5 accoridng should be according

Response:

Thanks for your careful checks. We are sorry for our carelessness. In our resubmitted manuscript, we have corrected the "accoridng" into "according". Thank you again for your reminder.

5. Page 6 first assembly should be combining HiFi and ONT not HiC and ONT since it's the unphased assembly

Response:

Thank you for pointing out the error. We have corrected this error in the revised manuscript. Your suggestion really means a lot to us.

6. Page 7 "nearly 90% coverage of for three data types", 90% of what? I assume this means that 90% of the assembly is covered by all three datatypes with the expected coverage but this is not clear from the current text.

Response:

In response to your comment, we have revised the text to clearly of our description on the coverage of the assembly by different data types. We appreciate your guidance in improving the clarity and precision of our manuscript.

7. Figure 1 in the review responses shows a likely indel on chr8 hap2. Is this pre-polishing? Was this polished with the authors' method?

Response:

Thank you for your careful examination of Figure 1 in the review responses, highlighting the indel observed on chr8 hap2. The indel identified was detected in the pre-polishing stage of our assembly process. Following this, we employed polishing method, which leverages both long-read and short-read sequencing data, to address and correct this indel. We appreciate your attention to this detail.

Response Fig. 1. IGV depiction of HiFi and UL reads mapping to the Hap2 genome across the gap region of Chr.8.

Above is the coverage depth and read mapping situation of HiFi data at gap upstream and downstream positions; below is the coverage depth and read mapping situation of UL reads.

Reviewer #3 (Remarks to the Author):

The revised manuscript has addressed all my concerns.

1. In line 415-416, "adaptive molecular response" is confusing. Although the author may not want

to express the response of immune system in that description, I need to say that there is only innate immune system in plant. Please change your description.

Response:

Thank you for your valuable comments. we apologize for the confusion caused by our description, and we totally agree with you. As you have pointed out, plants only possess innate immunity system, and deficiencies in acquired immunity (adaptive immunity). Our description of “adaptive molecular response” in the text, especially the word “adaptive”, can easily lead to ambiguity or misunderstanding about the existence of adaptive immunity in plants. So according to your suggestion, in the newly revised manuscript, we have removed the confusing description of "adaptive molecular response" and revised the relevant content. Thank you again for your valuable suggestions to improve the quality of our manuscript.

Reviewers' Comments:

Reviewer #1:

Remarks to the Author:

The authors have fully addressed my concerns. I have no additional comments.

Reviewer #2:

Remarks to the Author:

The authors have addressed my concerns, thank you for the detailed responses.

Reviewer #1 (Remarks to the Author):

The authors have fully addressed my concerns. I have no additional comments.

Response:

Thank you very much!

Reviewer #2 (Remarks to the Author):

The authors have addressed my concerns, thank you for the detailed responses.

Response:

Thank you very much!